# EqCollide: Equivariant and Collision-Aware Deformable Objects Neural Simulator

## Abstract

Simulating collisions of deformable objects is a fundamental yet challenging task due to the complexity of modeling solid mechanics and multi-body interactions. Existing data-driven methods often suffer from lack of equivariance to physical symmetries, inadequate handling of collisions, and limited scalability. Here we introduce EqCollide, the first end-to-end equivariant neural fields simulator for deformable objects and their collisions. We propose an equivariant encoder to map object geometry and velocity into latent control points. A subsequent equivariant Graph Neural Network-based Neural Ordinary Differential Equation models the interactions among control points via collision-aware message passing. To reconstruct velocity fields, we query a neural field conditioned on control point features, enabling continuous and resolution-independent motion predictions. Experimental results on 2D and 3D scenarios show that EqCollide achieves accurate, stable, and scalable simulations across diverse object configurations. It achieves $24.34\%$ to $57.62\%$ lower rollout MSE even compared with the best-performing baseline model. Furthermore, EqCollide could generalize to more colliding objects and extended temporal horizons, and stay robust to input transformed with group action.

## 1 Introduction

Simulating deformable object dynamics, particularly involving multiple bodies' collision, is essential for applications in computer graphics, digital twins, and robotics. Traditional physics-based numerical approaches, such as Finite Element Analysis (FEA) (Belytschko et al., 2014), Material Point Method (MPM) (Stomakhin et al., 2013), and Smoothed Particle Hydrodynamics (SPH) (Bender & Koschier, 2015), have provided solid frameworks to describe the dynamics of deformable objects under diverse conditions. Grounded on fundamental physical laws, these methods achieve high accuracy in capturing complex interactions between bodies. However, they often suffer from prohibitive computational costs.

Recently, Machine Learning (ML) has been increasingly applied to simulate collisions of deformable objects, aiming to overcome the limitations of classical physics-based methods. Broadly, these techniques could be categorized into data-driven approaches that learn directly from data and physics-informed approaches that integrate ML within the physical laws. A significant advancement in data-driven methods was the adoption of Graph Neural Networks (GNNs), which deliver a flexible and scalable framework for modeling systems with complex geometry and dynamical interactions (Li et al., 2019; Pfaff et al., 2020; Sanchez-Gonzalez et al., 2020; Cao et al., 2023). On the other hand, Physics-informed methods, such as Physics-Informed Neural Networks (PINNs) (Raissi et al., 2019), Deep Energy Method (DEM) (Samaniego et al., 2020), and Lagrangian neural networks (Cranmer et al., 2020), incorporate Partial Differential Equation (PDE) based priors via strong or weak formulations to integrate data-driven flexibility with physical consistency.

Despite the advances, critical challenges remain. **Deformable objects exhibit complex internal forces, and collisions introduce multi-body interactions, making it difficult for existing data-driven methods to accurately simulate their dynamics. A further challenge lies in ensuring the equivariance in latent representations of data-driven methods, which is essential for guaranteeing that predictions respect physical symmetries.** In the absence of equivariant embeddings, models may fail to generalize under transformations such as translations and rotations, resulting in physically implausible simulations. Several GNN-based simulators achieve equivariant embeddings

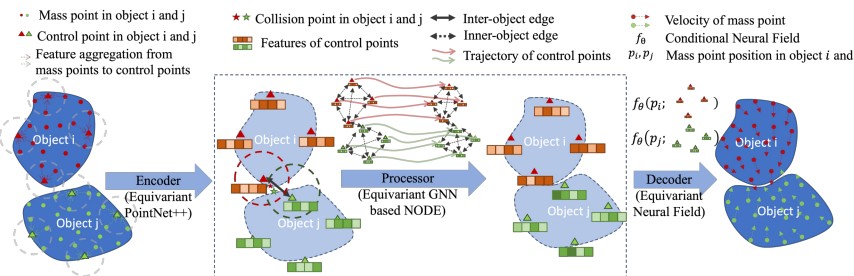

Figure 1: An encoder maps mass point states to control points via PointNet++; a processor (Equivariant GNN-based NODE) predicts dynamics; a decoder reconstructs the velocity field using an equivariant neural field conditioned on control point features.

by updating node and edge features with kernels derived from invariant encodings (Brandstetter et al., 2022; Satorras et al., 2021; Han et al., 2022; Huang et al., 2022; Valencia et al., 2024). While promising, these approaches have primarily focused on physical systems such as cloth, rigid bodies, and fluids, where the interactions are relatively simpler and do not involve the complex contact dynamics and deformations present in deformable-body collisions. Moreover, they typically work on systems discretized with a large number of nodes, resulting in high computational costs.

To address the challenges, we introduce Equivariant and Collision-Aware Deformable Objects Neural Simulator (EqCollide), a novel simulator that explicitly models collision for deformable objects. By embedding the states of deformable objects into a small set of control points with latent features and modeling their interactions through a GNN in a collision-aware manner, we achieve an efficient and scalable simulation for the complex dynamics of deformable objects during collision. As shown in Figure 1, EqCollide adopts an Encoder–Processor–Decoder architecture. We make a novel use of PointNet++ (Qi et al., 2017) by adapting it to encode position and velocity information of mass points into a compact set of control points. An equivariant GNN-based Neural Ordinary Differential Equation (NODE) (Chen et al., 2018) is then employed to simulate the dynamics of control points' positions and contexts via collision-aware message passing. To further tackle the challenge of maintaining equivariance in latent representations, EqCollide integrates $\mathbf{SE}(n)$-equivariance throughout the entire simulation pipeline. Specifically, we adapt PointNet++ to be $\mathbf{SE}(n)$-equivariant, ensuring consistent encoding of mass points into control points. The dynamics are then modeled using an equivariant NODE, and finally, an equivariant neural field conditioned on the control points' features reconstructs the velocity field for arbitrary mass point positions. As illustrated in Table 1, EqCollide is the first equivariant simulator that simultaneously achieves high computational efficiency, continuous field representation, and collision-aware interaction modeling. The contributions of EqCollide include:

**(1) End-to-End Equivariant Simulator:** We propose an end-to-end equivariant framework, where the Encoder, Processor, and Decoder are all designed to be equivariant. It guarantees that the entire simulation pipeline remains consistent under global transformations. Such equivariant properties also enable modular use of the model, as intermediate representations remain valid and meaningful regardless of global transformations applied to the input. **(2) Collision-Aware Massage Passing in Control Points:** A collision-aware graph is built on control points, where message passing between objects is selectively activated upon collision detection, ensuring efficient and physically consistent modeling of complex interactions both within and between objects. **(3) High-Fidelity Deformable Objects Collision Simulation:** Our method EqCollide achieves accurate, smooth, and resolution-independent prediction of velocity fields. Experimental results show that EqCollide achieves 24.34% to 57.62% lower rollout MSE even compared with the best-performing baseline model. It also shows improved simulation fidelity and generalization to longer prediction horizons and unseen scenarios.

Table 1: Comparison of different GNN-based models for deformable object simulation

| Model | Equivariance | Continuous Representation | Graph Size | Collision Detection |
|---|---|---|---|---|
| MeshGraphNets (Pfaff et al., 2020) | No | No | Large | Yes |
| GNS (Sanchez-Gonzalez et al., 2020) | No | No | Large | Yes |
| SGNN (Han et al., 2022) | Yes | No | Large | Yes |
| GMN (Huang et al., 2022) | Yes | No | Large | Yes |
| ENF-PDE (Knigge et al., 2024) | Semi-* | Yes | Small | No |
| EqCollide (Ours) | Yes | Yes | Small | Yes |

*Note: ENF-PDE obtains the latent using meta-learning that involves applying stochastic gradient descent to an initial latent. Although this step does not guarantee equivariance, the network architecture is equivariant to the latent.

## 2 RELATED WORK

### 2.1 IMPLICIT NEURAL REPRESENTATION

Implicit Neural Representations (INRs) model data as continuous functions parameterized by neural networks, mapping coordinates to relevant values. This paradigm has achieved significant progress across various applications, from 3D shape representation using Signed Distance Functions (Park et al., 2019; Chen & Zhang, 2019; Mescheder et al., 2019) to novel view synthesis via radiance fields (Mildenhall et al., 2020). Recent work has integrated geometric principles into INRs through frameworks like Equivariant Neural Fields (ENF) (Wessels et al., 2025), which bridges equivariant networks with neural fields. Subsequent extensions such as ENF-PDE (Knigge et al., 2024) and ENF2ENF (Catalani et al., 2025) have applied this framework to PDE problems and airfoil dynamics. However, these approaches rely on autodecoding optimization, resulting in input processing that lacks true equivariance. In contrast, our proposed method is end-to-end equivariant.

### 2.2 GNN-BASED SIMULATION

Modeling interactions between elements is a key challenge in simulating system dynamics. Graph Neural Networks (GNNs) have shown promise in physical simulation, with approaches like Mesh-GraphNets (Pfaff et al., 2020) establishing a paradigm for iterative state updates. Recent equivariant GNNs, such as Subequivariant Graph Neural Networks (SGNNs) (Han et al., 2022), have incorporated symmetry principles for improved generalization. Further advances include physics-informed approaches integrating Hamiltonian/Lagrangian mechanics (Zhong et al., 2021; Sanchez-Gonzalez et al., 2019) and geometric constraints (Huang et al., 2022).

However, most GNN-based simulators operate directly in physical space, leading to large graph sizes and a focus on rigid body collisions with simple interactions. In contrast, our proposed method operates in latent space, learning dynamics from a sparse set of control points through an end-to-end equivariant architecture, specifically targeting deformable body collisions with complex nonlinear patterns. Please refer to Appendix B for a detailed discussion of related work.

## 3 METHOD

We propose EqCollide, a graph-based, learnable simulator designed for collision-aware dynamics modeling. The simulator predicts a trajectory of system states $S_{0:T} = (S_0, S_1, ..., S_T)$ over $T$ time steps. As depicted in Figure 1, EqCollide comprises three main stages: an Encoder $\mathcal{E}$, a Processor $\mathcal{P}$, and a Decoder $\mathcal{D}$. The Encoder $\mathcal{E}$ maps the input point cloud $S_t$ at time $t$ to $N$ control points in the latent space. Each control point $z_t^i$ for $i = 0, ..., N-1$ is represented[1] by a tuple $\{p_t^{ctl}, c_t^{ctl}\}^i = \{\boldsymbol{x}_t^{ctl}, \alpha_t^{ctl}, c_t^{ctl}\}^i$, consisting of a 2D position $\boldsymbol{x}_t^{ctl} \in \mathbb{R}^2$, an orientation $\alpha_t^{ctl} \in \mathbb{R}$, and context information $c_t^{ctl} \in \mathbb{R}^c$, where superscript $ctl$ is abbreviated for control point. The Processor $\mathcal{P}$ is a NODE $F_\psi(z_t) = \frac{dz_t}{dt}$ that models the continuous temporal evolution of the latent system states. The Decoder $\mathcal{D}$ is an equivariant neural network $f_\theta$ that reconstructs the velocity field from the latent state, defined as $\hat{\boldsymbol{v}}_t(\boldsymbol{x}_t) = f_\theta(\boldsymbol{x}_t; z_t)$. The $\theta$ and $\psi$ denote learnable parameters of $\mathcal{E}$ and $\mathcal{D}$, respectively. The final positions of mass points are obtained by applying Euler integration to the reconstructed velocity field. For brevity, the method mainly uses the 2D example for illustration. It can be directly extended to 3D by generalizing position and velocity representations from 2D to 3D.

The remainder of this section is organized as follows. We first recall essential preliminaries on equivariance and the associated group actions used throughout this work. We then introduce the proposed Equivariant Encoder-Decoder architecture in Sec 3.1, which serves as the foundation of our end-to-end equivariant simulation framework. Next, we present a collision-aware GNN-based NODE module that models the temporal evolution of the system in Sec 3.2. Finally, we analyze the end-to-end equivariance property of EqCollide and introduce the overall training pipeline in Sec 3.3.

---

[1]The subscript $t$ denotes the time step, and the superscript $i$ indicates the $i^{\text{th}}$ control point.

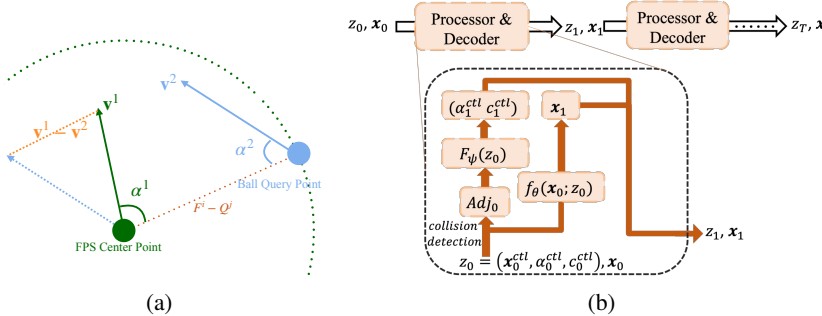

(a)            (b)

Figure 2: (a) Our rotation-invariants: [ $\alpha^1, \alpha^2, \|\mathbf{v}^1 - \mathbf{v}^2\|_2^2, \|F^i - Q^j\|_2^2, \cos(F^i - Q^j, \mathbf{v}^1 - \mathbf{v}^2)$ ]. These 5 invariants remain unchanged if we rotate the whole point cloud input. (b) Control points updating mechanism. Orientation and context of control points are updated with GNN $F_\psi$. While their positions $\boldsymbol{x}_t^{ctl}(t = 0, 1, \dots)$ are updated with $f_\theta$ as they are selected from $\boldsymbol{x}_t(t = 0, 1, \dots)$ instead of directly updated with $F_\psi$.

### 3.1 EQUIVARIANT ENCODER AND DECODER

Before introducing the model, we recall some preliminaries on the Special Euclidean group $\mathbf{SE}(n)$ and the group actions used in this paper. $\mathbf{SE}(n)$ consists of all rigid transformations (*i.e.*, rotations and translations) in an $n$-dimensional Euclidean space. We also consider its subgroup $\mathbb{R}^n$, which includes only translations. Equivariance is considered both in the physical space $\mathbb{R}^n$, where the deformable objects reside, and in the latent space $\mathcal{G}$, where their abstract representations evolve.

Formally, let $G$ denote a transformation group such that $G \subseteq \mathbf{SE}(n)$, and let $g \in G$ represent a specific group action. We denote the state of an object with $z_t^i = \{p_t^{ctl}, c_t^{ctl}\}^i (i = 1, ..., M)$. Under the action of $g$, the transformed state is defined as $g(p, c) = (gp, c)$ , which applies the transformation $g$ only to the pose $p$, while keeping the feature $c$ invariant. That is, $c$ is invariant under group actions.

We describe the state of the system with $N$ objects at time $t$ with $S_t = \{O_t^1, O_t^2, ..., O_t^N\}$. Each object is described with a Lagrangian representation, where multiple mass points are used to represent the movement of the whole object. State of the $i^{\text{th}}$ mass point is described with $(\boldsymbol{x}_t^i, \boldsymbol{v}_t^i) \in \mathbb{R}^4$, where $\boldsymbol{x}_t^i \in \mathbb{R}^2$ represents mass point positions and $\boldsymbol{v}_t^i \in \mathbb{R}^2$ represents mass point velocities.

Our proposed end-to-end equivariant framework starts with an equivariant Encoder, $\mathcal{E}$. This encoder takes input point clouds $\boldsymbol{x}_t$ and their velocities $\boldsymbol{v}_t$ at timestep $t$, and outputs a latent representation $z_t = \mathcal{E}(\boldsymbol{x}_t, \boldsymbol{v}_t)$. For any group action $g \in G$, our encoder is supposed to satisfy the equivariance property: $\mathcal{E}(g(\boldsymbol{x}_0, \boldsymbol{v}_0)) = g\mathcal{E}(\boldsymbol{x}_0, \boldsymbol{v}_0) = gz_0 = (gp_0, c_0)$.

Our framework is designed to achieve equivariance across different group actions. For pure translations, we use PointNet++ (Qi et al., 2017) and adapt the Farthest Point Sampling (FPS) method. Specifically, the initial seed point is chosen as the point farthest from the point cloud's centroid, rather than randomly. This particular initialization makes FPS operations equivariant under a translation group $G \subseteq \mathbb{R}(n)$. Furthermore, PointNet++'s approach of selecting control points directly from the input mass points naturally ensures that initially selected control points lie within the object.

Considering the rotation-translation group $\mathbf{SE}(n)$. The standard relative position vector $F^i - Q^j$ is not rotation-invariant, which precludes its direct use in $\mathbf{SE}(n)$ - equivariant aggregation, where $F_i, Q_j$ represent the $i^{\text{th}}$ FPS center point and its $j^{\text{th}}$ Ball query point, respectively. Therefore, we have developed specific rotation-invariant features tailored for the aggregation process when handling $\mathbf{SE}(n)$. Appendix C.2 contains further details.

Specifically, assume a FPS point $F^i$ has velocity $\boldsymbol{v}^1$ and a query point $Q^j$ within its neighborhood has velocity $\boldsymbol{v}^2$. We define five rotation-invariant scalars derived from these points and velocities: 1) The angle $\alpha^1$ between $\boldsymbol{v}^1$ and the relative position $(Q^j - F^i)$. 2) The angle $\alpha^2$ between $\boldsymbol{v}_2$ and the relative position $(F^i - Q^j)$. 3) The squared distance between the points: $\|F^i - Q^j\|_2^2$. 4) The squared magnitude of the velocity difference: $\|\boldsymbol{v}^1 - \boldsymbol{v}^2\|_2^2$. 5) The cosine similarity between the relative position and velocity difference: $\cos(F^i - Q^j, \boldsymbol{v}^1 - \boldsymbol{v}^2)$. Using these five scalar invariants as the input features for the PointNet aggregation module, instead of the raw relative position vector $F^i - Q^j$, ensures rotation invariance. See Figure 2a for an illustration.

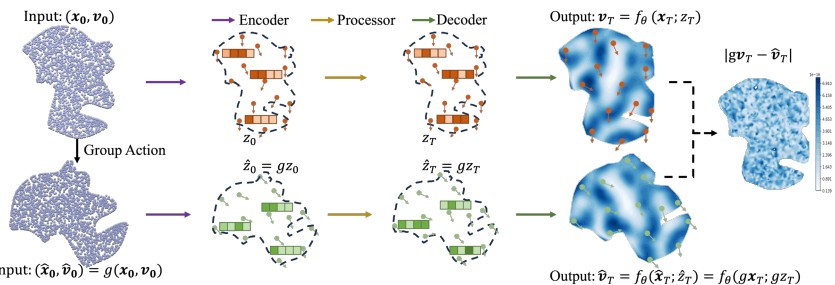

Figure 3: EqCollide achieves end-to-end equivariance, a property not attainable by autodecoder-based methods: when the input point cloud is transformed, the velocity field also transforms accordingly. EqCollide can be either $\mathbb{R}^2$ or $\mathbf{SE}(2)$ equivariant, depending on users' configuration.

Our decoder employs an implicit representation based on the ENF framework (Wessels et al., 2025): Given coordinates $x$ and the latent code $z$ from the encoder, it outputs the velocity field $v(x_t) = f_\theta(x_t; z)$. However, for representing $\mathbf{SE}(2)$ equivariant vector fields, the bi-invariant property mentioned in (Wessels et al., 2025) is unsuitable for vectors. The original $\mathbf{SE}(2)$-equivariant decoder $f_\theta$ produces rotation-invariant outputs, which contradicts the requirement that velocity fields should transform covariantly with the input. To overcome this limitation, we design a new decoder that ensures equivariance: $f_\theta(gx, gz) = g f_\theta(x, z)$. This guarantees that predicted velocities transform consistently under group actions, preserving the geometric properties of the physical system. The detailed construction of this decoder is provided in Appendix C.5.3.

## 3.2 COLLISION-AWARE GNN AS PROCESSOR

We propose an innovative $\mathbf{SE}(n)$-equivariant GNN-based NODE to model the dynamics of control points generated by the Encoder $\mathcal{E}$ for the perception of collision. We adopt PONITA (Bekkers et al., 2024) as the backbone of our processor. As illustrated in Figure 2b, our proposed collision-aware GNN-based NODE models predicts the derivatives of different variables in control point features $z$ separately. Specifically, the temporal derivatives of orientation $\alpha^{ctl}$ and context $c^{ctl}$ of control points are updated through the GNN processor: $(\alpha_t^{ctl}, c_t^{ctl}) = (\alpha_{t-1}^{ctl}, c_{t-1}^{ctl}) + F_\psi(z_{t-1}) \cdot dt$. The derivative of control points' position $x^{ctl}$ is computed separately using a conditional neural field; the updates at each time step are defined as $x_t^{ctl} = x_{t-1}^{ctl} + f_\theta(x_{t-1}^{ctl}; z_{t-1}) \cdot dt$. All mass point positions are also updated using the same neural field $f_\theta$ according to the update rule: $x_t = x_{t-1} + f_\theta(x_{t-1}; z_{t-1}) \cdot dt$. As a result, the updated control points can be directly retrieved from the updated mass point. These indices of points, through which control points are selected from mass points, are determined at the beginning of the trajectory by the encoder described in Section 3.1. Since indices of control points remain fixed in each trajectory, the motion of control points remains inherently coupled with the motion of the corresponding objects. This design ensures consistent physical grounding and facilitates accurate collision awareness during the entire simulation process. In contrast, ENF-PDE updates the positions of control points and mass points independently. This decoupling can lead to control points drifting away from the objects they are meant to represent, compromising the physical fidelity of the simulation.

Another key advantage of our approach is the incorporation of collision-aware message passing, which enables physically grounded interactions between control points. Control points from different objects pass messages only when a collision is detected between objects. Please refer to Appendix C.3 for details on the collision-aware message passing.

## 3.3 END-TO-END EQUIVARIANT FRAMEWORK

As introduced in Section 3.1, the Encoder in EqCollide is equivariant to the group action $g \in G \subseteq \mathbf{SE}(n)$. Besides, the equivariance of the Processor is elaborated in Appendix A, and the equivariance of the Decoder (ENF), has already been proved in Section 3.1, preserving that $\mathcal{D}(gx; gz) = g(\mathcal{D}(x; z))$.

After all, we made a theoretical guarantee that the equivariance during the overall Encoder-Processor-Decoder process ensures better geometrical properties when modeling the complex dynamical system.

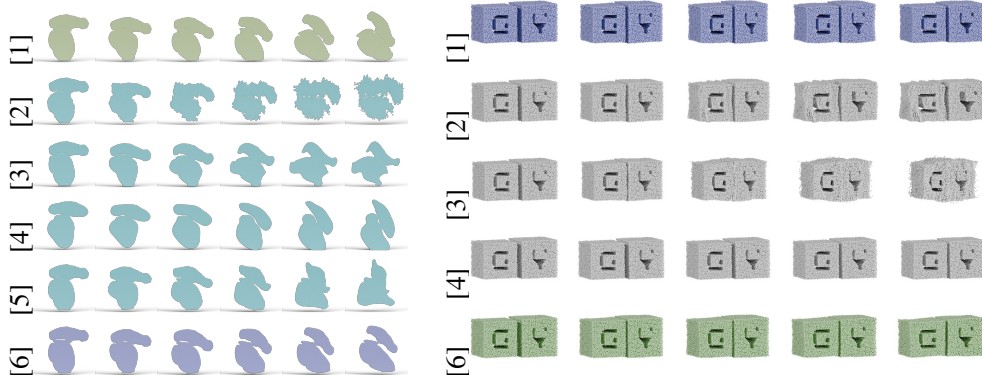

(a) 2D Collision Visualization Results.     (b) 3D Collision Visualization Results.

Figure 4: Visualization for rollout prediction testing results of EqCollide and baseline models: [1]. Ground Truth, [2]. MGN (Pfaff et al., 2020), [3]. SGNN (Han et al., 2022), [4]. ENF(SDF) (Knigge et al., 2024), [5]. ENF(Vel) (Knigge et al., 2024) [6]. EqCollide). Our visualization shows that graph-based methods struggle with deformable-object collisions due to discrete representations. ENF(SDF) methods face issues with sharp geometry. ENF displacement fields have poor elasticity and lack end-to-end equivariance.

$$g\mathcal{D}(\boldsymbol{x}_t; \mathcal{P}(\mathcal{E}(\boldsymbol{x}_0, \boldsymbol{v}_0))) = \mathcal{D}(g\boldsymbol{x}_t; \mathcal{P}(\mathcal{E}(g(\boldsymbol{x}_0, \boldsymbol{v}_0)))) \tag{1}$$

To verify the end-to-end equivariance of our framework, we conduct a validation test as follows. Given an input point cloud with or without group action, we evaluate two processing pipelines: (1) first apply a group action to the input point cloud, then process it with the simulator to obtain the output field; (2) first input the original point cloud into the simulator to generate the field, then apply the corresponding group action to obtain final result. The test result shown in Figure 3 illustrates that our model could output exactly equivariant results under group action, demonstrating that the end-to-end equivariance described by Equation 1 holds for our entire framework.

**Loss Function.** We proposed a loss function that consists of both displacement prediction loss $\mathcal{L}_{\text{dis}}$ and reconstruction loss $\mathcal{L}_{\text{recons}}$ as below to train the overall EqCollide.

$$\mathcal{L} = c_1\mathcal{L}_{\text{dis}} + c_2\mathcal{L}_{\text{recons}}, \quad \mathcal{L}_{\text{dis}} = \frac{1}{T}\sum_{t=0}^{T}\|\boldsymbol{x}_t - \hat{\boldsymbol{x}}_t\|^2, \mathcal{L}_{\text{recons}} = \frac{1}{T}\sum_{t=0}^{T}\|\boldsymbol{v}_t - f_\theta(\boldsymbol{x}_t; z_t)\|^2 \tag{2}$$

where $\hat{\boldsymbol{x}}_t$ is the predicted positions from EqCollide, and $c_1$, $c_2$ are weighting coefficients for two loss terms. We adopt a two-stage training strategy to optimize the framework efficiently. In the first stage, only the reconstruction loss $\mathcal{L}_{\text{recons}}$ is used to train the model, allowing the neural field decoder $f_\theta$ and the Encoder $\mathcal{E}$ to learn accurate velocity reconstruction from ground-truth trajectories. In the second stage, all components, including $\mathcal{E}$, $F_\psi$, $f_\theta$, are jointly optimized using the full objective defined in Equation 2. Both $c_1$ and $c_2$ are set to 1. The training procedure is elaborated in Appendix C.4.

## 4 EXPERIMENT

### 4.1 EQCOLLIDE IMPLEMENTATION SETTINGS

To evaluate our proposed framework's capability in learning complex dynamics, we introduce a challenging dataset, **DeformableObjectsCollision**, which consists of 4 different collision scenarios for 2D and 3D deformable objects. This dataset provides a unique benchmark for evaluating the scalability, generalization, and physical fidelity of learning-based deformable dynamics simulators.

We compare EqCollide against three types of baselines: ENF-PDE trained with SDF field and velocity field data, respectively, and two well-established graph-based simulators, MeshGraphNets (Pfaff et al., 2020) and SGNN (Han et al., 2022). We include both SDF and velocity field variants of ENF-PDE to evaluate its capability in the collision task. Notably, ENF-PDE employs meta-learning for generating control points and represents the field in an Eulerian perspective, contrasting with our Lagrangian

approach. MeshGraphNets and SGNN both represent interactions using dense, node-based graphs in the physical space. Details of the dataset and baselines are available in Appendix C.1 and C.6.

## 4.2 EXPERIMENTAL RESULTS IN 2D SCENARIO

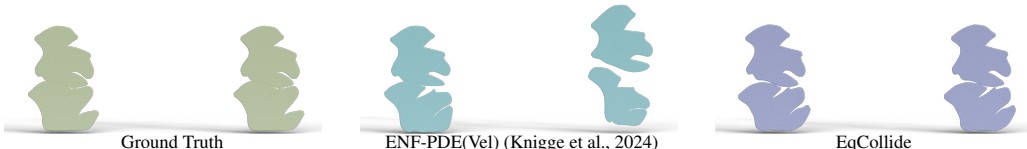

Ground Truth        ENF-PDE(Vel) (Knigge et al., 2024)        EqCollide

Figure 5: End-to-end equivariance application in downstream tasks. Empirical evaluations show that our method accurately recovers transformed states, even when tested in coordinates outside the training distribution. Unlike the semi-equivariant ENF-PDE method, our framework exhibits equivariance to the input state. The experiments show that our method can accurately capture the transformed state, even when evaluating in an unseen coordinate system during training.

In 2D collision scenario simulation, we employ the mean squared error (MSE) between the predicted and ground truth positions as the evaluation metric. The deformation predicted by EqCollide and baseline models is visualized in Figure 4a. Table 2 shows that EqCollide consistently achieves the lowest prediction MSE across both evaluation settings: unseen combinations of seen objects and entirely unseen object shapes. Specifically, compared with the strongest baseline model ENF-PDE (Vel), EqCollide reduces MSE by $35.82\%$ on unseen object combinations and $24.34\%$ on unseen object shapes after 25 steps. For a shorter rollout of 20 steps, this reduction further increases to $66.89\%$ on unseen object combinations and $32.44\%$ on unseen object shapes. We attribute these superior results to our dedicated geometry-grounded framework and collision-aware message passing, as detailed in the ablation study. The table also indicates that all models exhibit slightly higher prediction errors on unseen shapes compared to unseen object combinations, highlighting the greater generalization challenge posed by novel geometries. As expected, MSE increases with rollout steps for all models due to the accumulation of errors over time.

Table 2: Displacement prediction MSE in unseen object combinations and unseen shapes.

| Prediction | MSE in unseen object combination($\times 10^{-6}$) ↓ | | | | | | MSE in unseen object shape($\times 10^{-6}$) ↓ | | | | | |
|---|---|---|---|---|---|---|---|---|---|---|---|---|
| scenario | 1-step | 5-step | 10-step | 15-step | 20-step | 25-step | 1-step | 5-step | 10-step | 15-step | 20-step | 25-step |
| ENF-PDE (Vel) | 0.309 | 2.767 | 9.780 | 18.909 | 30.806 | 48.300 | 0.293 | 2.715 | 13.133 | 32.666 | 63.205 | 113.395 |
| MeshGraphNets | 1.980 | 20.100 | 79.000 | 229.000 | 581.000 | 1230.000 | 2.452 | 23.621 | 93.017 | 251.589 | 593.489 | 1233.300 |
| SGNN | 0.224 | 7.130 | 32.100 | 84.900 | 186.000 | 332.000 | 0.226 | 8.203 | 41.958 | 114.697 | 238.362 | 413.729 |
| EqCollide | **0.003** | **1.270** | **4.600** | **7.780** | **11.200** | **31.000** | **0.002** | **1.880** | **9.220** | **21.800** | **42.700** | **85.800** |
| EqCollide-$\mathbf{SE}(n)$ | 0.280 | 4.490 | 12.300 | 22.100 | 32.900 | 55.300 | 0.302 | 4.380 | 16.100 | 34.800 | 61.500 | 106.933 |

Note: In this comparison, ENF-PDE (Vel) refers to the ENF-PDE model trained with velocity field data. EqCollide-$\mathbf{SE}(n)$ denotes the model with SE(n) equivariance, while EqCollide corresponds to the model with $\mathbb{R}^2$ equivariance. This table doesn't include ENF-PDE (SDF) because the SDF representation does not provide explicit trajectories for mass points. However, the performance of ENF-PDE (SDF) is qualitatively compared with other models in the visualization results.

Interestingly, the $\mathbf{SE}(n)$-equivariant variant of EqCollide doesn't demonstrably outperform its $\mathbb{R}^2$-based counterpart or the strongest baseline ENF-PDE (Vel) in terms of prediction MSE on testing samples. We hypothesize that this is attributable to the significantly increased constraints imposed by enforcing strict rotation equivariance, which can complicate the optimization process and potentially reduce the model's capacity for certain complex dynamics. This observation aligns with prior findings (Wessels et al., 2025; Deng et al., 2021) suggesting that stricter geometric constraints can sometimes impede model performance.

This phenomenon highlights an inherent trade-off between achieving improved geometric consistency and optimization complexity. While our end-to-end equivariant EqCollide may exhibit slightly lower average accuracy, it provides a strong guarantee of robustness under group-transformed inputs, as illustrated in Figure 5. Such inherent geometric consistency is crucial as it can simplify out-of-distribution downstream tasks and potentially reduce the need for extensive training data. In contrast, standard non/semi-equivariant models like ENF-PDE typically exhibit significant performance degradation when evaluated on spatially transformed inputs. As a result, even though the fully

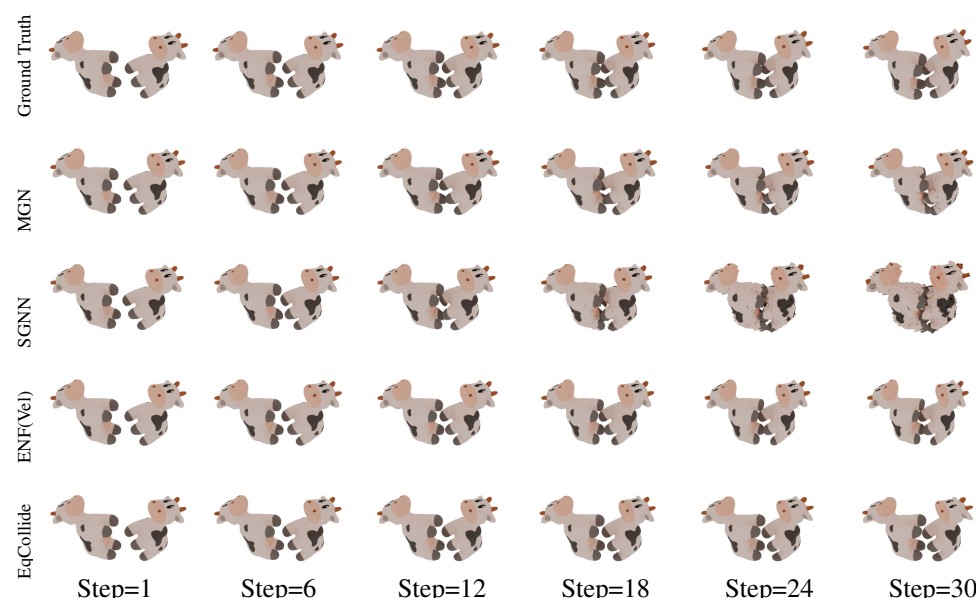

Figure 6: Visualization results of EqCollide and baseline models on spot cow model. MGN and SGNN suffer from severe mesh self-intersections. Furthermore, while ENF-PDE may appear comparable in individual snapshots, its temporal deformation behavior deviates more significantly from the ground truth. Conversely, our method demonstrates superior adherence to the Ground Truth and generates more realistic deformations.

SE(n)-equivariant variant may not always beat a non-equivariant baseline on displacement prediction accuracy, its geometric guarantees, better physical interpretability, and robustness to transformations remain central and meaningful contributions.

Figure 4a shows that the movement and deformation predicted by EqCollide are visually closer to the ground truth compared with baseline models. MeshGraphNets and SGNN can simulate the initial stage of free fall but diverge much faster than other models when objects collide. The ENF-PDE with SDF can generate the overall movements of objects, but the shape of objects is overly smoothed, especially with increasing rollout steps. This might be because of the limited high-frequency representation ability of SDF. The ENF-PDE with velocity field exhibits the best visualization performance among baseline models. However, compared to EqCollide, it exhibits more cases of penetration, which might result from the lack of constraints on the movement of control points and the absence of collision detection. Another 2D sample from the test set with unseen shapes is included in Appendix D.1.

Table 3: MSE of EqCollide without end-to-end equivariance and collision-aware message passing

| Prediction | MSE in unseen object combination($\times 10^{-6}$) ↓ | | | | | | MSE in unseen object shape($\times 10^{-6}$) ↓ | | | | | |
|---|---|---|---|---|---|---|---|---|---|---|---|---|
| scenario | 1-step | 5-step | 10-step | 15-step | 20-step | 25-step | 1-step | 5-step | 10-step | 15-step | 20-step | 25-step |
| Ablation-1 | 0.234 | 4.230 | 12.900 | 22.900 | 39.400 | 77.000 | 0.202 | 3.700 | 14.600 | 34.800 | 68.300 | 124.672 |
| Ablation-2 | 0.299 | 3.407 | 15.118 | 55.096 | 159.927 | 357.739 | 0.314 | 3.497 | 18.963 | 63.356 | 162.845 | 346.695 |

Note: Ablation-1 denotes the EqCollide without any equivariance. Ablation-2 denotes the EqCollide without collision-aware message passing.

## 4.3 EXPERIMENTAL RESULTS IN 3D SCENARIOS

We evaluate EqCollide and baseline models through simulating three 3D collision scenarios and use MSE as the evaluation metric. We measured rollout errors for all models at various timesteps in the same test set. The errors for the two 3D cubic and one 3D cow-shaped deformable body collision scenarios are summarized in Table 4 and Table 5, respectively.

In the cubic 3D deformable body collision on horizontal plane, EqCollide yields the lowest prediction error when the rollout step exceeds 5. In the 3D collision in midair, EqCollide yields the lowest

Table 4: Displacement prediction MSE in cubic 3D deformable body collision

| Prediction | MSE for 3D collision on horizontal plane ($\times 10^{-6}$) $\downarrow$ | | | | | | MSE for 3D collision in midair ($\times 10^{-6}$) $\downarrow$ | | | | | |
| scenario | 1-step | 5-step | 10-step | 15-step | 20-step | 25-step | 1-step | 5-step | 10-step | 15-step | 20-step | 25-step |
|---|---|---|---|---|---|---|---|---|---|---|---|---|
| ENF-PDE (Vel) | 0.2225 | 6.0146 | 27.4560 | 62.0669 | 104.2109 | 147.9833 | 1.3936 | 13.7729 | 32.8807 | 45.9916 | 52.9889 | 54.5098 |
| MeshGraphNets | 0.0011 | 0.0206 | 0.2097 | 1.0089 | 3.1167 | 7.2937 | 0.0464 | 0.6175 | 3.0004 | 8.5146 | 17.2402 | 27.3320 |
| SGNN | **0.0002** | 0.0127 | 0.2404 | 1.4954 | 4.7324 | 10.7509 | **0.0013** | **0.1067** | **1.1311** | **5.6375** | 19.8267 | 52.1498 |
| EqCollide | 0.0003 | **0.0054** | **0.0484** | **0.2931** | **1.0772** | **3.0912** | 0.0075 | 0.1480 | 1.6577 | 6.1762 | **12.5160** | **17.9152** |

prediction error when the rollout step exceeds 15. Compared to the best baseline, EqCollide achieves 57.62% and 34.45% reduction on MSE at 25th steps for the two 3D scenarios, respectively. It is worth noting that although SGNN attains smaller error in several rollout steps in both 3D scenarios, EqCollide also achieves small and comparable error in those steps. Moreover, as shown in Figure 4b, EqCollide produces collision process with much higher fidelity over all baselines in simulating both 3D scenarios. In contrast, other baselines exhibit distortions that contradict our physical intuition. We observe that although ENF-PDE yields relatively large MSE, its visualization results appear satisfactory. This is because the visualization indicates that the model fails to capture the deformation of the blocks and primarily simulates only their translational motion. Full visualization results for the two 3D cubic deformable body collision scenarios are provided in Appendix D.2.

We also conducted a collision experiment involving 3D cow-shaped deformable body collision. We trained EqCollide and the baselines on the same dataset and tested the results at novel angles and velocities. Table 5 presents the numerical results. The overall performance is consistent with our findings in the previous two 3D collision scenarios. EqCollide achieves the lowest rollout prediction error when the rollout horizon exceeds 5 steps. Notably, while SGNN performs better in the initial 5 time steps, it demonstrates poor capability in long-term rollouts. We believe it results from SGNN's lack of spatially continuous representation and inefficient message passing due to the large graph size. Although ENF-PDE appears visually similar to our method in Figure 6, the quantitative data confirms that our method achieves the best performance in long-term rollouts.

These results demonstrate that our method is well-suited to 3D deformable objects collision tasks, maintaining superior properties and outperforming all baseline models. Furthermore, the properties of equivariance, clear physical interpretability, and expressive efficiency, which have been previously validated in 2D scenarios, remain intact in 3D settings.

Table 5: Displacement prediction MSE in Spot cow 3D deformable body collision

| Prediction | MSE for 3D cow-shaped deformable body collision ($\times 10^{-6}$) $\downarrow$ | | | | | |
| scenario | 1-step | 5-step | 10-step | 15-step | 20-step | 25-step |
|---|---|---|---|---|---|---|
| ENF-PDE (Vel) | 0.0071 | 0.1125 | 0.5881 | 1.7226 | 3.8218 | 7.2454 |
| MeshGraphNets | 0.0414 | 0.4874 | 1.9395 | 4.8680 | 10.0842 | 18.7400 |
| SGNN | **0.0008** | **0.0513** | 0.5133 | 2.0614 | 5.5260 | 11.8415 |
| EqCollide | 0.0034 | 0.0931 | **0.4873** | **1.3516** | **2.8159** | **4.9163** |

## 4.4 Ablation Study

We conduct two ablation experiments to validate the contribution of end-to-end equivariance and collision-aware message passing in our proposed EqCollide. In Ablation-1, we disrupt the end-to-end equivariance of EqCollide through replacing bi-invariants for $\mathbf{SE}(n)$ with the sum of features for each edge. In ENF-PDE, the equivariance is achieved by constructing an invariant attribute, $p^{-1}x$, the relative distance in a Lie Group, which is defined in (Knigge et al., 2024; Bekkers et al., 2024; Wessels et al., 2025). Here we replace all bi-invariants with $x + p$, which destroys the equivariance property. In Ablation-2, we evaluate the impact of collision-aware message passing. Instead of using dynamic collision detection to define the graph connections, we employ a static adjacency matrix that only connects control points belonging to the same object. This effectively disables inter-object message passing. The prediction errors from both ablation experiments are summarized in Table 3. Compared with the full EqCollide model presented in Table 2, both ablated versions exhibit significantly higher MSE across different rollout steps and test sets. These results underscore the importance of both end-to-end equivariance and our collision-aware design for achieving accurate and robust deformable object simulation.

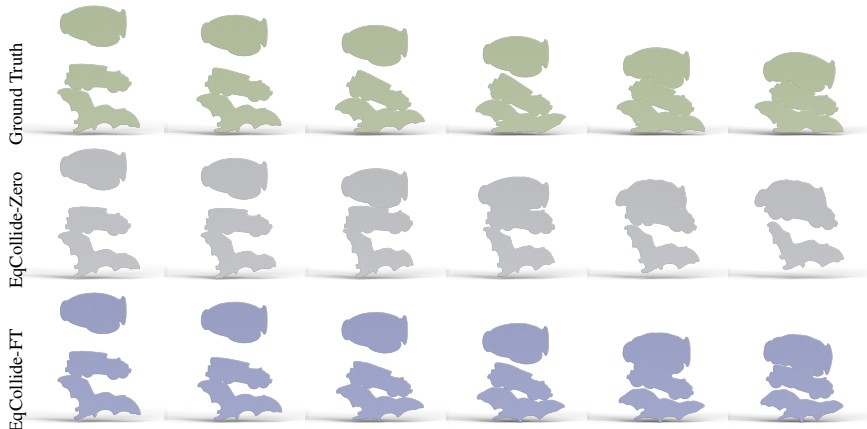

Figure 7: Visualization of EqCollide on three objects collision. EqCollide-Zero means the model was not finetuned and used for zero-shot inference. EqCollide-FT represents the finetuned model.

## 4.5 SCALABILITY STUDY

We evaluate EqCollide's scalability with respect to increasing object interactions and longer prediction horizons. First, we assess EqCollide on a challenging three-object collision scenario unseen in training data. This setup requires the model to generalize learned two-object collision patterns to more complex interactions. As depicted in the first two rows of Figure 7, the pretrained EqCollide can qualitatively simulate the deformation of all three objects upon contact. However, the predicted displacements exhibit larger visual discrepancies compared to two-object scenarios, indicating limited accuracy without adaptation. To enhance performance, we finetune EqCollide using an additional dataset comprising only 20% of the training set involving three-object collisions. After 300 epochs of finetuning, the model's prediction accuracy improves significantly. As illustrated in Figure 7, the simulated trajectories exhibit realistic collision behaviors that closely resemble the ground truth. Furthermore, we evaluate EqCollide's generalization ability on four-object interactions. After finetuning on a small set of four-object collision samples, EqCollide is able to generate visually plausible trajectories. This demonstrates EqCollide's strong potential to generalize to more complex scenarios with minimal adaptation. For further finetuning results and exploration of scalability, please refer to Appendix D.3.

We also investigate temporal scalability by extending the rollout length up to 50 steps. We compare the rollout prediction MSE of EqCollide with baselines over rollout steps ranging from 1 to 50. As shown in Appendix D.4, baselines exhibit rapidly increasing errors due to cumulative prediction error, while EqCollide consistently maintains lower prediction errors across the entire time horizon, validating its robustness in long-horizon simulation tasks. Although EqCollide outperforms baselines, it still accumulates errors during long-horizon rollouts. Furthermore, its prediction accuracy on complex scenarios like three-object collisions still requires further improvement.

## 5 CONCLUSION

We have presented EqCollide, an end-to-end equivariant and collision-aware framework for simulating the collision dynamics of deformable objects. EqCollide enforces equivariance throughout the pipeline to ensure consistency under global transformations and incorporates a collision-aware message passing mechanism to embed physical priors. Experiments show that EqCollide consistently produces trajectories closely aligned with ground truth across unseen object combinations and shapes. It also scales effectively with respect to rollout length and object count, requiring only minimal finetuning. Moreover, we demonstrate that end-to-end equivariance enables robust generalization to out-of-distribution inputs under geometric transformations. By enforcing end-to-end equivariance, EqCollide guarantees transformation consistency, thereby enabling modular integration into broader simulation pipelines, which we believe could be a promising direction in GNN-based simulators. As future work, we plan to extend EqCollide to scenarios where more deformable bodies interact simultaneously and to a broader range of rigid-body collision benchmarks. Generalization ability across different material properties also represents an important direction for future work.

ETHICS STATEMENT

All authors have read and agreed to adhere to the ICLR Code of Ethics at https://iclr.cc/public/CodeOfEthics. We confirm that this work complies with all general ethical principles outlined therein.

REPRODUCIBILITY STATEMENT

All model code, datasets, and rollout videos are provided in https://anonymous.4open.science/r/EqCollide-F3E9/. Please refer to Appendix C.1 for more complete description of the dataset and Appendix C.5 for details on model training.

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

APPENDIX

## A    PROOF OF EQUIVARIANT PROCESSOR

As introduced in Section 3.2, we utilize a GNN-based NODE $F_\psi(z)$ in the PROCESSOR to model the evolution of control points' orientation and context features in latent space, defined as follows:

$$(\alpha_{t+\delta t}^{ctl}, c_{t+\delta t}^{ctl}) = \mathcal{I}_\psi(\alpha_t^{ctl}, c_t^{ctl}) = \int_t^{t+\delta t} F_\psi(z_t)dt + (\alpha_t^{ctl}, c_t^{ctl}), \tag{3}$$

where $\mathcal{I}_\psi$ denotes the integration operator applied over $F_\psi$.

In this section, we first prove that $F_\psi$ is equivariant under any group action, followed by a proof that $\mathcal{I}_\psi$ preserves equivariance.

$F_\psi$ is built upon the PONITA architecture used in (Knigge et al., 2024), which employs convolutional weight-sharing over bi-invariants and has been theoretically proved to be equivariant. The main differences between $F_\psi$ and PONITA lie in the construction of the adjacency matrix and the design of the convolution kernels. Specifically, in $F_\psi$:

- The adjacency matrix Adj encodes interactions between control points $z$ and is dynamically updated based on collision detection.

- Distinct convolution kernels $k_{\text{inter}}$ and $k_{\text{inner}}$ are used for inter-object and intra-object message passing.

Importantly, the resulting Adj remains invariant under group actions, as it depends solely on the relative distances between mass points and control points, which are themselves invariant under rotation and translation. Likewise, the use of separate kernels $k_{\text{inter}}$ and $k_{\text{inner}}$ doesn't break equivariance, as both operate on invariants within their respective equivalence classes and preserve the symmetry structure under $\mathbf{SE}(n)$ transformations.

Therefore, the enhancements introduced in $F_\psi$ maintain the invariance property as below:

$$\forall g \in G, \begin{cases} \text{Adj}(z) = \text{Adj}(gz) \\ k_{\text{inner}}(z) = k_{\text{inner}}(gz) \\ k_{\text{inter}}(z) = k_{\text{inter}}(gz) \end{cases}. \tag{4}$$

The integration operator $\mathcal{I}_\psi$ can be shown to be equivariant under any group action $g \in G$ as follows:

$$\begin{aligned} \mathcal{I}_\psi(g(\alpha_t^{ctl}, c_t^{ctl})) &= \int_t^{t+\delta t} F_\psi(gz_t)dt + (g\alpha_t^{ctl}, c_t^{ctl}) \\ &= \int_t^{t+\delta t} gF_\psi(z_t)dt + g(\alpha_t^{ctl}, c_t^{ctl}) \\ &= g(\int_t^{t+\delta t} F_\psi(z_t)dt + (\alpha_t^{ctl}, c_t^{ctl})) \\ &= g\mathcal{I}_\psi(\alpha_t^{ctl}, c_t^{ctl}) \end{aligned} \tag{5}$$

Moreover, it should be noted that $x$ is updated using Decoder $\mathcal{D}$ in EqCollide as illustrated in Section 3.2. Our proposed Decoder $\mathcal{D}$ has been shown to be equivariant, i.e., $\forall g \in G, gf_\theta(x^{ctl}) = f_\theta(gx^{ctl})$ in Section 3.1. The control points' position $x^{ctl}$ is updated via the integration operator $\mathcal{I}_\theta$:

$$\mathcal{I}_\theta(x_t^{ctl}) = \int_t^{t+\delta t} f_\theta(x_t^{ctl})dt + x_t^{ctl} = x_{t+\delta t}^{ctl} \tag{6}$$

Analogously to Equation 5, $\mathcal{I}_\theta$ can be proved to be equivariant to any group action $g \in G$ as follows:

$$
\begin{aligned}
\mathcal{I}_\theta(g\boldsymbol{x}_t^{ctl}) &= \int_t^{t+\delta t} f_\theta(g\boldsymbol{x}_t^{ctl})dt + g\boldsymbol{x}_t^{ctl} \\
&= \int_t^{t+\delta t} gf_\theta(\boldsymbol{x}_t^{ctl})dt + g\boldsymbol{x}_t^{ctl} \\
&= g\mathcal{I}_\theta(\boldsymbol{x}_t^{ctl})
\end{aligned} \tag{7}
$$

Combining the equivariant updates of $\boldsymbol{x}^{ctl}$, $\alpha^{ctl}$, and $c^{ctl}$, the full latent state $z_t = (\boldsymbol{x}_t^{ctl}, \alpha_t^{ctl}, c_t^{ctl})$ is updated as follows:

$$
\begin{aligned}
\mathcal{I}(gz_t) &= (\mathcal{I}_\theta(g\boldsymbol{x}_t^{ctl}), \mathcal{I}_\psi(g\alpha_t^{ctl}, c_t^{ctl})) \\
&= g(\mathcal{I}_\theta(\boldsymbol{x}_t^{ctl}), \mathcal{I}_\psi(\alpha_t^{ctl}, c_t^{ctl})) \\
&= g\mathcal{I}(z_t),
\end{aligned} \tag{8}
$$

thereby proving that the entire Processor to update latent states is equivariant under any group action $g \in G$.

# B    EXTENDED RELATED WORK

## B.1    IMPLICIT NEURAL REPRESENTATION

Architectural improvements include SIREN's use of sinusoidal activations for high-frequency details (Sitzmann et al., 2020b). Training strategies encompass autodecoders (Park et al., 2019) and meta-learning methods like MetaSDF (Sitzmann et al., 2020a). INRs have also been successfully applied to image data (Dupont et al., 2022) and various physics simulations, including elasticity (Chen et al., 2023) and fluid dynamics (Deng et al., 2023; Tao et al., 2024).

ENF-PDE specifically integrates the PONITA architecture (Bekkers et al., 2024) within the INR backbone for solving PDE problems, while ENF2ENF focuses on aerodynamic applications. The autodecoding limitation affects how these methods process input coordinate fields, compromising true geometric equivariance.

## B.2    GNN-BASED SIMULATION

GNNs have been extensively applied to physical simulation (Battaglia et al., 2016; Kipf et al., 2018; Sanchez-Gonzalez et al., 2018), with Graph Network Simulator (GNS) (Sanchez-Gonzalez et al., 2020) and MeshGraphNets (Pfaff et al., 2020) forming the basis for multi-rigid-body collision modeling (Allen et al., 2023; Rubanova et al., 2024).

Various methods have integrated physical priors, including Hamiltonian/Lagrangian mechanics (Zhong et al., 2021; Sanchez-Gonzalez et al., 2019) and geometrical constraints (Huang et al., 2022). The broader landscape of equivariant GNNs includes (Satorras et al., 2021; Thomas et al., 2018; Brandstetter et al., 2022; Gasteiger et al., 2021; Kofinas et al., 2024; Bekkers et al., 2024; Deng et al., 2021), with SGNNs (Han et al., 2022) utilizing hierarchical graphs for information propagation.

# C    IMPLEMENTATION DETAILS

## C.1    DATASET CONSTRUCTION DETAILS

To create the four collision scenarios in **DeformableObjectsCollision** dataset, we simulate a wide range of deformable object trajectories using the Taichi-MPM simulator (Hu et al., 2019). These collision scenarios in **DeformableObjectsCollision** serve as severely challenging benchmarks and are significantly more demanding than those found in particle-based fluid simulations or rigid-body systems. Because they require models to capture internal material continuity and complex interactions between deformable objects.

Our experimental datasets cover different boundary conditions and material to reflect different physical setups. The 2D collision scenario comprises simulations of diverse 2D deformable objects

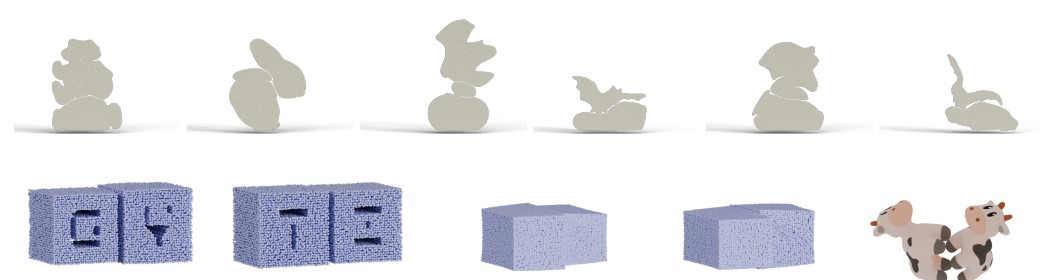

Figure 8: Examples from Collision Scenarios in **DeformableObjectsCollision**.

falling under gravity and colliding with the ground or other bodies. The objects are initialized with randomized orientations and dropped from different heights under a constant gravitational field. All object shapes are sampled from the 2D Shape Structure Dataset (Carlier et al., 2016), with MIT License Copyright (c) [2016]. The Young's modulus and Poisson ratio of our simulated objects are 2000 Pa and 0.4. As shown in Figure 8, the 2D collision scenario includes a variety of different shapes, such as cars and bottles. For 2D scenarios, we generate 1,000 training trajectories, 50 validation trajectories, and 100 test trajectories. In test trajectories, 50 of them are novel objects, and 50 of them consist of novel combinations with seen objects in the training set. Each trajectory consists of 60 timesteps, and each timestep corresponds to 0.002 seconds in the physical world. To balance rollout prediction accuracy and training efficiency, each collision trajectory is divided into multiple sequences of 20 time steps. EqCollide is trained to predict the positions of the subsequent steps in each sequence based on the position and velocity of the objects at the first time step. Every object in a trajectory is composed of 10,000 mass points, making this a large-scale and high-resolution simulation benchmark.

The three 3D collision scenarios comprise the collision of two 3D deformable objects under different physical conditions. In the first 3D scenario, two objects slide without friction on a horizontal plane and collide with each other. Objects were cubes of identical volume but with random shapes, initial positions, and velocities. In the second scenario, two cubical blocks with alphabet-shaped cutouts from Thingi10K Dataset (Zhou & Jacobson, 2016) are initialized with varying side lengths. One block is assigned varying initial velocities across trajectories and collides with the other block in midair in free space without gravity. The Young's modulus and Poisson ratio of our simulated 3D cubic objects are 300 Pa and 0.4. In the third scenario, two cow-shaped 3D deformable bodies collide in midair without gravity. In each sample, the two cow models collide at varying angles and initial velocities. The Young's modulus and Poisson ratio of the simulated 3D cow-shaped objects are 200 Pa and 0.4. For all 3D collision scenarios, we generated 900 training, 50 validation, and 50 test trajectories, where test trajectories featured unseen shapes or initial velocities. Each 40-timestep trajectory (1 timestep = 0.002s) was divided into 20-step sequences, with each object comprising 27,000 mass points. These 3D setups highlight deformable object interactions under planar contact and free-flight conditions, complementing the 2D deformable object collisions.

### C.2    EQUIVARIANT POINTNET++

In this section, we first introduce an $\mathbb{R}^n$-Equivariant PointNet++ and prove its equivariance. Then, we show how a $\mathbf{SE}(n)$-Equivariant PointNet++ can be naturally constructed by replacing the translation invariants in the $\mathbb{R}^n$-Equivariant version with invariants under any group action $g \in G$.

The Farthest Point Sampling (FPS) algorithm in PointNet++ (Qi et al., 2017) samples points from the input point cloud to achieve uniform spatial coverage. It begins by randomly selecting an initial point and then iteratively adds the point that is farthest from the current sampled set. However, this procedure is not equivariant to translation due to the randomness of the initial point selection. To ensure equivariance, we modify the FPS strategy by selecting the one that is farthest from the centroid of the entire point cloud as the initial point. This change guarantees both translation and rotation equivariance of FPS.

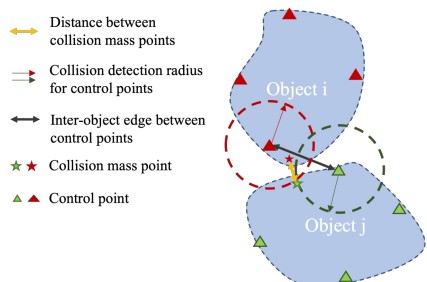

Figure 9: Collision detection strategy in GNN. Inter-object edges are added to control points only when collision of mass points is detected.

Following FPS, a Ball Query operation groups neighboring points within a predefined radius around each sampled point. For a given sampled point $F^i$ and its neighboring point $Q^j$ within the query ball, we compute translation-invariant features based on their relative positions. These invariants are processed by a PointNet++ module to aggregate local information. The resulting feature becomes the context vector associated with $F^i$, serving as input to the next layer. By repeating this Set Abstraction (SA) process (FPS, Ball Query, and PointNet++ aggregation) across multiple layers, we obtain a set of downsampled points (final FPS points) and their associated features. Under the translation group, this modified PointNet++ architecture inherently preserves translational equivariance.

In our deformable object simulation task, the input point cloud features include the initial positions $\boldsymbol{x}_0$ and velocities $\boldsymbol{v}_0$ of mass points. As shown in Equation 9, the $\mathbb{R}^n$-Equivariant PointNet++ is used to aggregate point cloud features into $M$ control points, each sampled from the mass points:

$$\mathcal{E}(\boldsymbol{x}_0, \boldsymbol{v}_0) = \{\boldsymbol{x}_0^{ctl}, \alpha_0^{ctl}, c_0^{ctl}\}^i = \{p_0^{ctl}, c_0^{ctl}\}^i = z_0^i, (i = 1, ..., M). \tag{9}$$

The control point positions $\boldsymbol{x}_0^{ctl}$ and orientations $\alpha_0^{ctl}$ are directly inherited from the position and velocity orientations of sampled mass points. As long as the sampling strategy is invariant under translation and rotation, these quantities are naturally equivariant under any group action. Each control point's context feature $c^i$ is computed by aggregating features from $K$ nearby mass points selected via Ball Query, based on a designed attribute $a^j$: $c^i = h_\theta(a^1, a^2, ..., a^K)$, where $h_\theta$ is a learnable message aggregation network. The attribute $a^j = F^i - Q^j$ is the relative position vector between the FPS point $F^i$ and its $j^{\text{th}}$ neighbor $Q^j$. Since relative position vectors are invariant to translation, the aggregated context $c^i$ is also translation-invariant.

Consequently, the Encoder is equivariant to translation:

$$\forall g \in G, g(\mathcal{E}(\boldsymbol{x}_0, \boldsymbol{v}_0)) = g\{p_0^{ctl}, c_0^{ctl}\}^i = \{gp_0^{ctl}, c_0^{ctl}\}^i = \mathcal{E}(g(\boldsymbol{x}_0, \boldsymbol{v}_0)), (i = 1, ..., M). \tag{10}$$

To construct an $\mathbf{SE}(n)$-Equivariant PointNet++, we replace the relative position vector $F^i - Q^j$ with a set of five invariants under group action $g \in G$ as introduced in Section 3.1. This substitution generalizes the encoder's equivariance from pure translation ($\mathbb{R}^n$) to the full special Euclidean group ($\mathbf{SE}(n)$). Since the argument is analogous to the previous proof for the $\mathbb{R}^n$-Equivariant case, we omit the details here.

### C.3    COLLISION-AWARE MESSAGE PASSING

As Figure 9 shows, a collision is considered to occur when the Euclidean distance between any two mass points falls below a threshold. To localize the interactions between control nodes, the detected collision must occur within a circular region centered at a control point. This mechanism ensures that only relevant control points participate in message passing, enabling EqCollide the perception of collision events and localized message passing patterns. Both the mass point collision threshold and the radius of the collision-aware region are treated as hyperparameters and set to 0.05 in all experiments.

### C.4    TRAINING PROCEDURE OF EQCOLLIDE

The two-stage training procedure of EqCollide is summarized in Algorithm 1.

---

**Algorithm 1** Training Procedure of EqCollide

---

**Require:** Mass point states $\{\boldsymbol{x}_t, \boldsymbol{v}_t\}, (t = 0, 1, ..., T)$
**Ensure:** Trained encoder $\mathcal{E}$ (PointNet++), NODE $F_\psi$, and neural field decoder $f_\theta$

1: **Stage 1: Velocity Field Reconstruction Pretraining**
2: Sample initial mass point positions and velocities from two objects
3: Encode mass points into control points: $z_t = \mathcal{E}(\boldsymbol{x}_t, \boldsymbol{v}_t)$
4: Reconstruct velocity field: $\hat{\boldsymbol{v}}_t = f_\theta(\boldsymbol{x}_t; z_t)$
5: Train $\mathcal{E}$ and $f_\theta$ using: $\mathcal{L}_{\text{recons}} = \frac{1}{T} \sum_{t=0}^{T} \|\boldsymbol{v}_t - f_\theta(\boldsymbol{x}_t; z_t)\|^2$
6: **End Stage 1**
7: **Stage 2: Joint Training of Encoder, Processor, and Decoder**
8: Initialize control point features: $z_0 = \mathcal{E}(\boldsymbol{x}_0, \boldsymbol{v}_0)$
9: Predict initial position: $\hat{\boldsymbol{x}}_1 = \boldsymbol{x}_0 + f_\theta(\boldsymbol{x}_0; z_0) \cdot dt$
10: **for** $t = 1$ to $T$ **do**
11:     Update control point orientation and context via NODE:$(\alpha_t^{ctl}, c_t^{ctl}) = (\alpha_{t-1}^{ctl}, c_{t-1}^{ctl}) + F_\psi(z_{t-1}) \cdot dt$
12:     Predict velocity: $\hat{\boldsymbol{v}}_{t-1} = f_\theta(\boldsymbol{x}_{t-1}; z_{t-1})$
13:     Update mass point positions: $\hat{\boldsymbol{x}}_t = \hat{\boldsymbol{x}}_{t-1} + \hat{\boldsymbol{v}}_{t-1} \cdot dt$
14:     Select updated control point positions $\hat{\boldsymbol{x}}_t^{ctl}$ from $\hat{\boldsymbol{x}}_t$
15:     Update control point features: $z_t = (\hat{\boldsymbol{x}}_t^{ctl}, \alpha_t^{ctl}, c_t^{ctl})$
16: **end for**
17: Compute total loss: $\mathcal{L} = c_1 \cdot \mathcal{L}_{\text{dis}} + c_2 \cdot \mathcal{L}_{\text{recons}}$
18: Update parameters of $\mathcal{E}$, $F_\psi$, and $f_\theta$ via backpropagation
19: **End Stage 2**

---

## C.5 NETWORK ARCHITECTURE

In 2D collision scenarios, we train all models for 2000 epochs using the Adam optimizer and cosine learning rate scheduler with an initial learning rate of $1 \times 10^{-3}$. In 3D collision scenarios, most training parameters match the 2D setup. To balance efficiency and accuracy, we trained for 200 epochs. We use a batch size of 16 and train on 4 NVIDIA H800 GPUs. Gradient clipping with a max norm of 1.0 is applied to ensure training stability. All quantitative metrics are computed with the test set.

### C.5.1 ENCODER ARCHITECTURE

Our Encoder network architecture primarily features a PointNet++ encoder, which leverages SA layers for multi-scale feature learning on point cloud data.

The PointNet++ (Qi et al., 2017) encoder consists of a series of stacked SA layers, designed to downsample the input point cloud and extract high-level features progressively. Each SA layer is a critical unit for point cloud feature extraction, incorporating a Multi-Layer Perceptron (MLP) for local feature encoding, and efficiently aggregating neighborhood information using FPS and ball querying operations. The operation is as follows:

FPS: The SA layer first selects a specified number of centroid points from the input point cloud. Unlike random selection, we choose the point furthest from the point cloud's centroid as the initial sample, then iteratively select points furthest from the chosen ones to ensure uniform coverage.

Ball Querying: For each selected centroid, the SA layer searches for the $max_k$ nearest neighbor points within a specified radius around it.

Local Feature Encoding: The SA layer calculates the relative coordinates of these neighbor points with respect to their centroids. If the input features include orientational information, we compute additional rotation-invariant features such as the magnitudes of relative coordinates and features, their dot product, and various angular differences. These features are then fed into an MLP for learning local patterns. Then the MLP's output is aggregated via a max pooling operation, which captures the most salient features within each local region, generating the final feature representation for each centroid.

The SA layer outputs the downsampled point cloud coordinates, the aggregated features, and the indices from farthest point sampling. Within the encoder, each SA operation also passes and updates velocity information from the previous layer to the next, further enhancing feature learning.

Specifically, the PointNet++ encoder is constructed with the following configurations: 1) MLP Dimensions List: [[32, 32, 64], [64, 64, 128], [128, 128, 32]]. These dimensions define the size of the MLP within each SA layer, progressively increasing the feature dimensionality. 2) Number of Samples List: [512, 128, 16] for 2D and [512, 128, 16] for 3D. The point cloud is successively downsampled from 512 points to 16 or 27 points, enabling feature extraction from coarse to fine granularity. 3) Radius List: [0.025, 0.05, 0.1]. The radius for neighborhood querying in each SA layer gradually increases, allowing deeper layers to capture broader contextual information. 4) Max Neighbors List: [32, 64, 128]. The maximum number of neighbor points considered in the ball query also incrementally increases with each SA layer.

Through this hierarchical and progressively abstractive approach, our PointNet++ encoder effectively extracts multi-scale, locally invariant features from raw point clouds, providing rich representations for subsequent tasks.

### C.5.2 PROCESSOR ARCHITECTURE

The processor network $F_\psi$ is designed based on the PONITA architecture (Bekkers et al., 2024), with modifications to incorporate a collision-aware adjacency matrix and separate convolutional kernels for handling inner-object and inter-object message passing. Besides, as mentioned in Section 3.2, the vectors update in NODE is with "hard constraint", which means the positions of control points are exactly set to the positions calculated with the velocity field obtained from the decoder; meanwhile, the scalars (specifically the orientations and context features) of control points are updated as follows:

The input node features are first embedded into a 64-dimensional latent space via a linear layer. Subsequently, three convolutional blocks set as message passing layers, with hidden dim set to 64 and widening factor set to 2, are applied using the collision-aware adjacency matrix to iteratively update the node representations in latent space. Within each message passing layer, messages are computed using two distinct sets of convolution kernels. One set of kernels works on generating a message for inner-object edges and the other for inter-object edges. These kernels are generated by encoding invariant features using two separate MLPs, each with a hidden dimension of 64. The polynomial degree of the basis is set to 3, and the dimension of the basis is set to 64. The messages are then aggregated from senders to receivers to update the corresponding node features. After three rounds of message passing, we use a single MLP layer as the readout layer to produce the target node features, which are the time derivatives of the orientations and context features. For more details, please refer to (Knigge et al., 2024).

### C.5.3 DECODER ARCHITECTURE

In our EqCollide model, we employ the ENF in (Wessels et al., 2025) as the decoder. This specific ENF implementation is configured for $\mathbf{SE}(2)$ equivariance. Functioning as a neural field, it takes 2D coordinates as input and produces 2D velocity as output. The decoder architecture incorporates 2 attention heads and a single self-attention layer. Its internal MLP has a hidden dimension of 64. For latent representation in 2D situation, the model utilizes 16 latent points, each defined by 2 coordinates and possessing 32 hidden dimensions; and in 3D situation, the model utilizes 27 latent points, each defined by 3 coordinates. A Gaussian window of 0.1 is consistently applied across all windows for these latent points. Furthermore, to effectively represent high-frequency data, we apply Random Fourier Feature (RFF) embedding. The standard deviation of the RFF for the Query (Q) components within the self-attention layer is set to 0.05, while for the Value (V) components, it is 0.2.

The original $\mathbf{SE}(2)$-equivariant decoder $f_\theta$ from (Wessels et al., 2025) satisfies $f_\theta\big(g\boldsymbol{x}, gz\big) = f_\theta\big(\boldsymbol{x}, z\big)$, where $g \in G \subseteq \mathbf{SE}(2)$ acts on poses in $z$. However, this formulation leads to rotation-invariant vector fields, which are physically implausible for velocity predictions since velocities should rotate with the object.

To achieve proper equivariance, we redesign the decoder to obey $f_\theta\big(g\boldsymbol{x}, gz\big) = g f_\theta\big(\boldsymbol{x}, z\big)$. Our solution constructs the equivariant decoder as:

$$f_\theta^{equiv}(\boldsymbol{x}\,;\,z) = \mathbf{R}\big(\phi_{\boldsymbol{x}}\big)\, f_\theta^{inv}(\boldsymbol{x}\,;\,z),$$

where $\mathbf{R}(\cdot)$ is the rotation matrix and $\phi_{\boldsymbol{x}}$ is an $\mathbf{SE}(2)$-equivariant orientation computed as the angle between the ray from the object's centroid to its farthest point and the positive x-axis. The function $f_{\theta}^{inv}$ produces invariant vector predictions, which are then rotated by $\mathbf{R}(\phi_{\boldsymbol{x}})$ to ensure proper transformation under group actions.

This construction guarantees that the decoder outputs transform covariantly with the input, making it suitable for predicting physically consistent velocity fields in deformable object simulations.

### C.6 BASELINE IMPLEMENTATION DETAILS

Two ENF-PDE based baseline models are used to model the collision dynamics of deformable bodies. Specifically, ENF-PDE (Vel) is trained to predict the velocity field based on the initial positions of mass points in objects. In contrast, ENF-PDE (SDF) is trained to estimate the SDF over point cloud data given the coordinates of sampled points. The hyperparameters in these two models exactly follow the original settings in (Knigge et al., 2024).

MeshGraphNets has demonstrated strong performance across a wide range of physical simulation tasks, while SGNN is specifically designed to handle object-collision scenarios and has been demonstrated to be effective in such settings.

The inputs to both MeshGraphNets and SGNN include the current node position, velocity, and vertical distances to the four scene boundaries. For MeshGraphNets, we additionally assign a categorical node attribute indicating object membership, allowing the model to distinguish between different objects. Consistent with the original paper (Pfaff et al., 2020), noises are added to input node positions and velocities to improve their robustness in rollout prediction. The model outputs node-wise acceleration, which is used to update both velocity and position through integration. In contrast, SGNN does not require explicit object identity labels in node attributes, as it separates inter-object and inner-object edges and processes them independently. The output of SGNN is the predicted velocity of each node. The design of SGNN used in this paper is consistent with the SGNN for RigidFall in the original paper (Han et al., 2022).

## D ADDITIONAL EXPERIMENTAL RESULTS

### D.1 2D VISUALIZATION RESULTS ON UNSEEN SHAPES

Figure 10 visualizes the rollout prediction results on another sample from the test set, featuring previously unseen object shapes. Consistent with our observation in Figure 5 of the main paper, movement and deformation predicted by EqCollide are visually closer to the ground truth compared with baseline models. MeshGraphNets and SGNN are capable of capturing the free fall at the initial several rollout steps. However, their predictions diverge significantly earlier than those of other models once collisions occur. ENF-PDE (SDF) is able to reproduce the overall motion trajectories of objects. Nevertheless, similar to our findings in Figure 5, it tends to produce overly smoothed object shapes. ENF-PDE (Vel) also achieves the most visually accurate results among the baseline models. But it tends to produce object shapes that deviate more significantly from the ground truth, exhibiting pronounced deformations.

### D.2 3D VISUALIZATION RESULTS

Figure 11 shows one set of visualization results for the collision of 3D cubical blocks with alphabet-shaped cutouts in midair. The Figures 12 and 13 show visualization results for the collision of 3D cubical blocks sliding on a horizontal plane.

### D.3 GENERALIZATION ABILITY TO MORE OBJECTS

We evaluate the generalization ability of EqCollide in more complex scenarios involving an increased number of interacting objects, and compare its performance against three representative baseline models. This experiment is designed to assess whether the learned dynamics can effectively scale to more challenging multi-body interactions without requiring model reconfiguration or architectural changes. To ensure a fair comparison, all models are fine-tuned on an identical dataset that contains

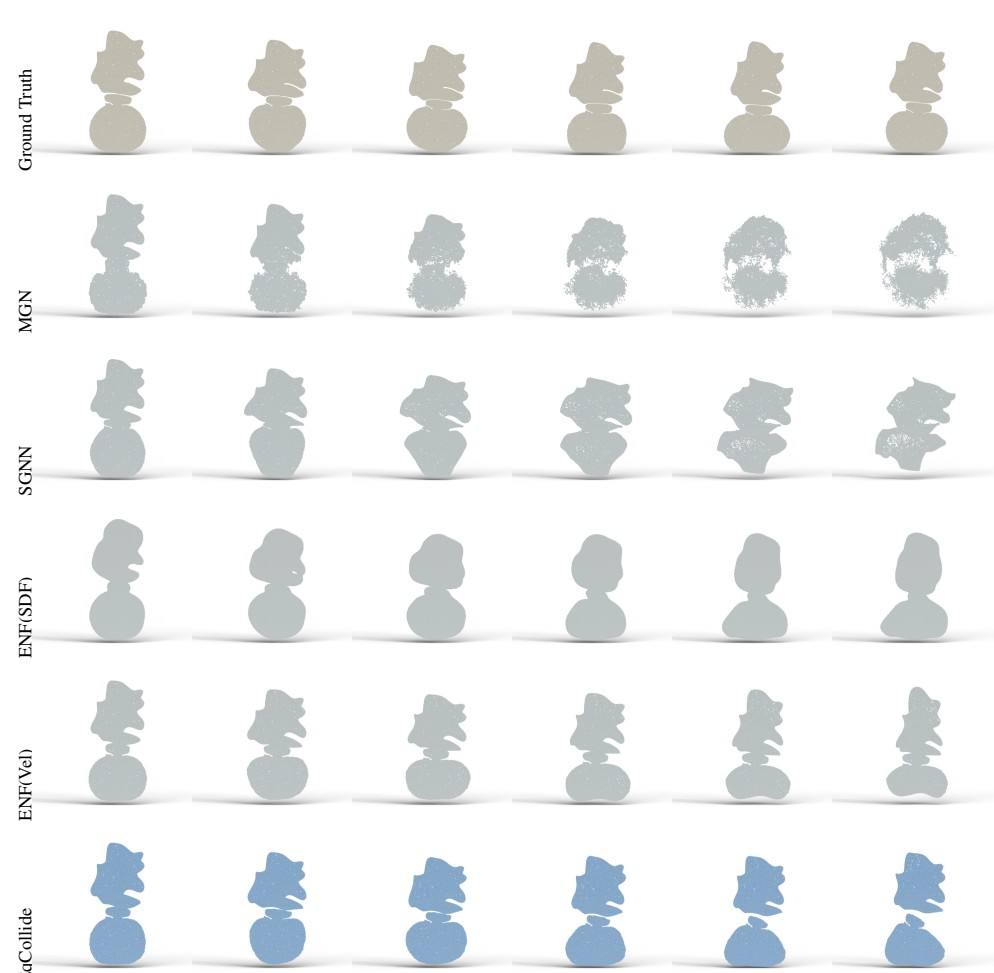

Figure 10: Visualization results of EqCollide and baseline models on unseen shapes. Each row in the figure shows six predicted frames from left to right, corresponding to rollout steps of 5, 10, 15, 20, 25, and 30. This convention is followed consistently across all visualizations throughout the paper.

collision trajectories involving three objects with varying shapes and initial positions. This dataset consists of 200 training samples and 50 testing samples. This setup introduces more intricate object-object interactions and a higher degree of physical complexity compared to the two-object training setting. Since all baseline models adopt a GNN-based processor to learn system dynamics, they are inherently generalizable to scenarios involving a larger number of objects. This is because GNNs operate on graph-structured data and can naturally scale to graphs with varying numbers of nodes, effectively allowing the models to simulate object collisions by simply increasing the number of nodes in the input graph.

The quantitative evaluation results are summarized in Table 6, where we report the MSE of displacement predictions at different rollout time steps. As shown in the table, EqCollide consistently achieves the lowest MSE across all rollout horizons, clearly outperforming the baselines. This indicates that EqCollide not only learns accurate short-term dynamics, but also maintains stability and physical plausibility in long-term predictions, even when extrapolating to more complex multi-object systems.

In addition to quantitative metrics, we also provide qualitative comparisons in Figure 15, where the predicted collision processes of all models on two samples from the test set are visualized. Both visual results further support our conclusions: predictions generated by EqCollide align more closely with the ground truth in both trajectory and object deformation, while baseline models tend to exhibit noticeable divergence or unnatural deformations as the simulation progresses.

To further assess the scalability and generalization ability of EqCollide, we additionally include a four-object collision experiment. After fine-tuning on a small set of four-object collision samples,

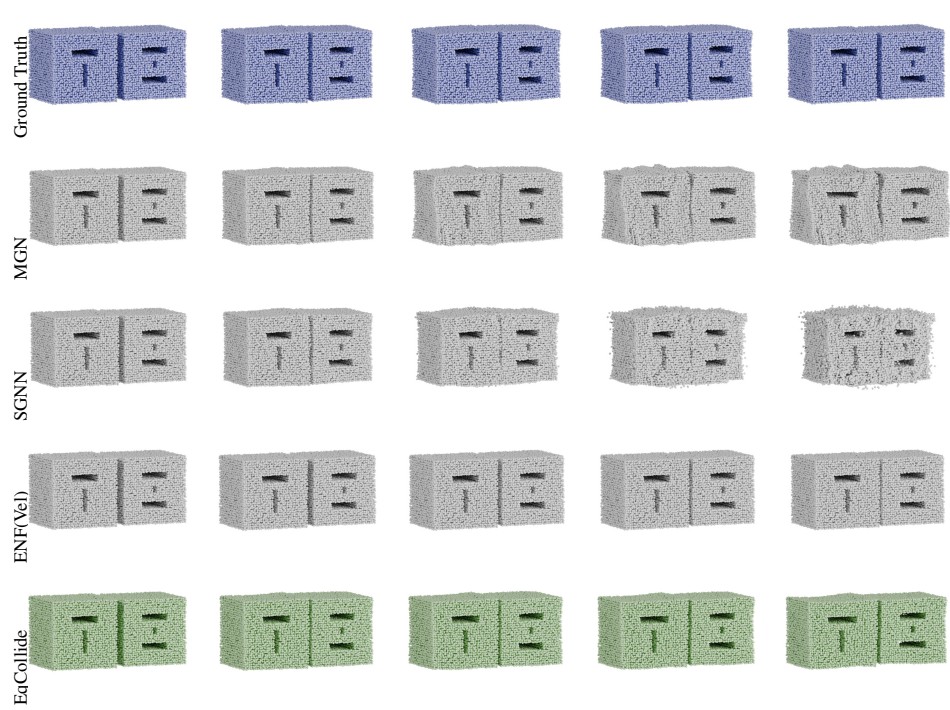

Figure 11: Visualization results of 3D alphabet block collision.

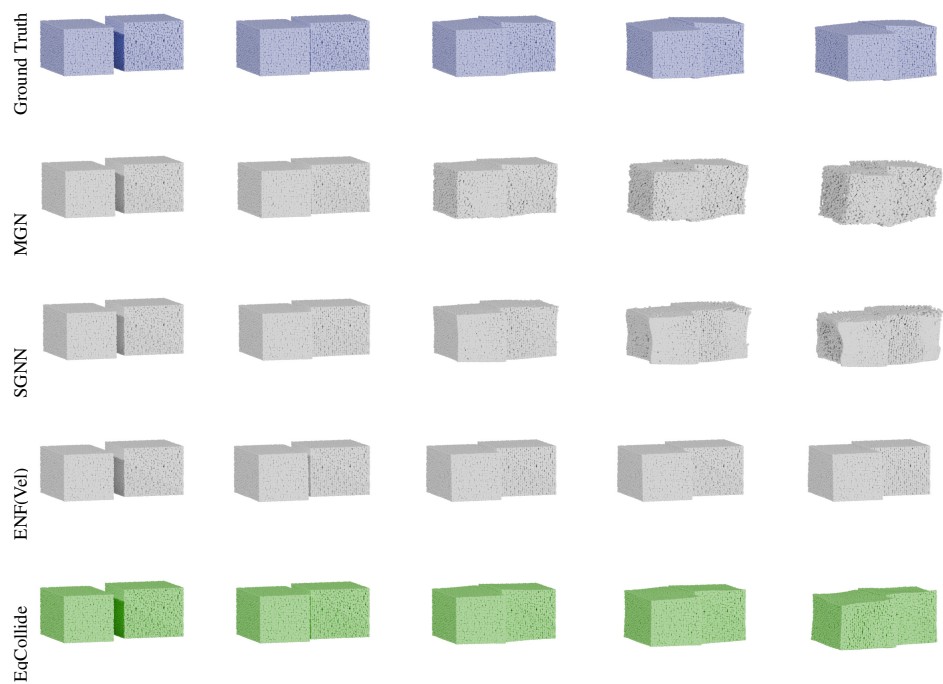

Figure 12: Visualization results of 3D cubic block collision.

EqCollide is able to produce visually plausible and stable rollouts. The displacement prediction MSE results at different rollout steps are reported in Table 7, and predicted trajectories are summarized in Figure 14. As illustrated in Figure 14, the zero-shot inference results exhibit larger visual discrepancies compared to two-object scenarios, which is consistent with the results shown in Figure 7. However, after finetuning, the predicted trajectories and deformations closely match the

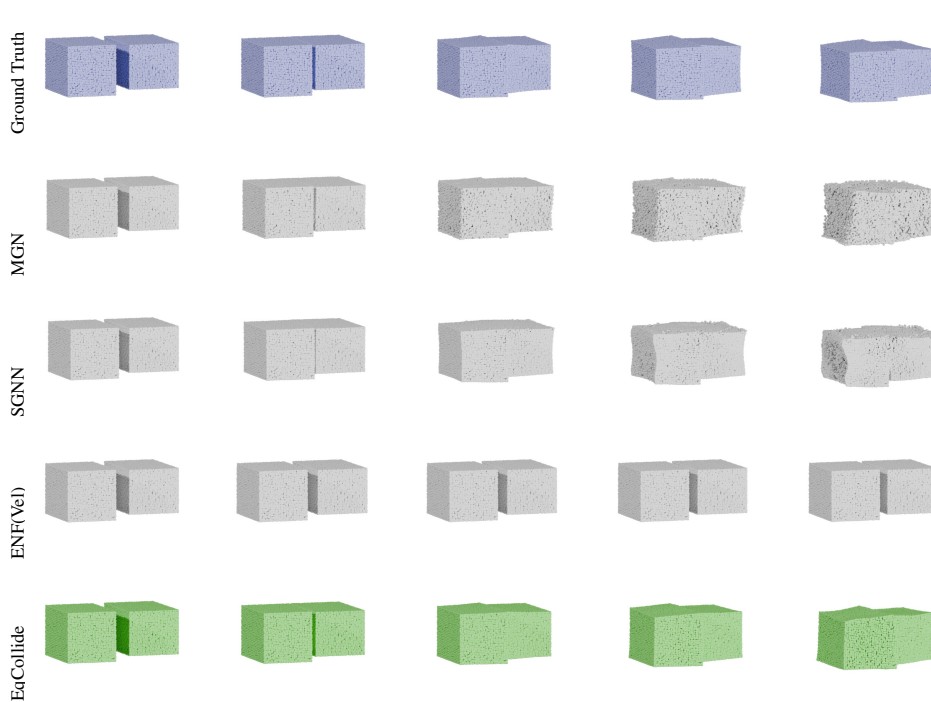

Figure 13: Visualization results of another 3D cubic block collision.

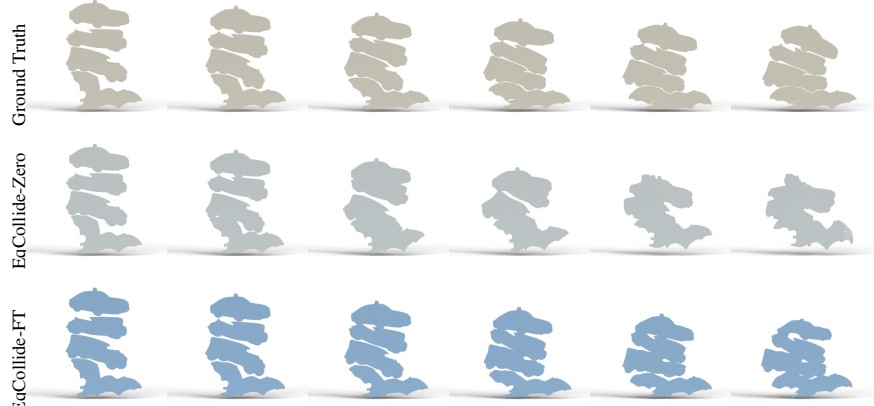

Figure 14: Visualization of EqCollide on four objects collision. EqCollide-Zero means the model was not finetuned and used for zero-shot inference. EqCollide-FT represents the finetuned model.

ground truth. This additional experiment further demonstrates that EqCollide can generalize beyond the training setting to more challenging scenarios involving a larger number of interacting deformable objects.

Table 6: MSE of EqCollide and baseline models after finetuning on three-object collision samples

| Prediction | Prediction error in unseen 3 objects($\times 10^{-6}$) $\downarrow$ | | | | | |
|---|---|---|---|---|---|---|
| scenario | 1-step | 5-step | 10-step | 15-step | 20-step | 25-step |
| ENF-PDE (Vel) | 1.407 | 8.299 | 19.188 | 33.354 | 50.061 | 76.318 |
| MeshGraphNets | 1.309 | 16.685 | 69.022 | 179.758 | 374.968 | 627.234 |
| SGNN | 0.231 | 6.454 | 33.590 | 83.165 | 218.990 | 618.453 |
| EqCollide | **0.002** | **1.299** | **5.882** | **11.192** | **17.680** | **34.092** |

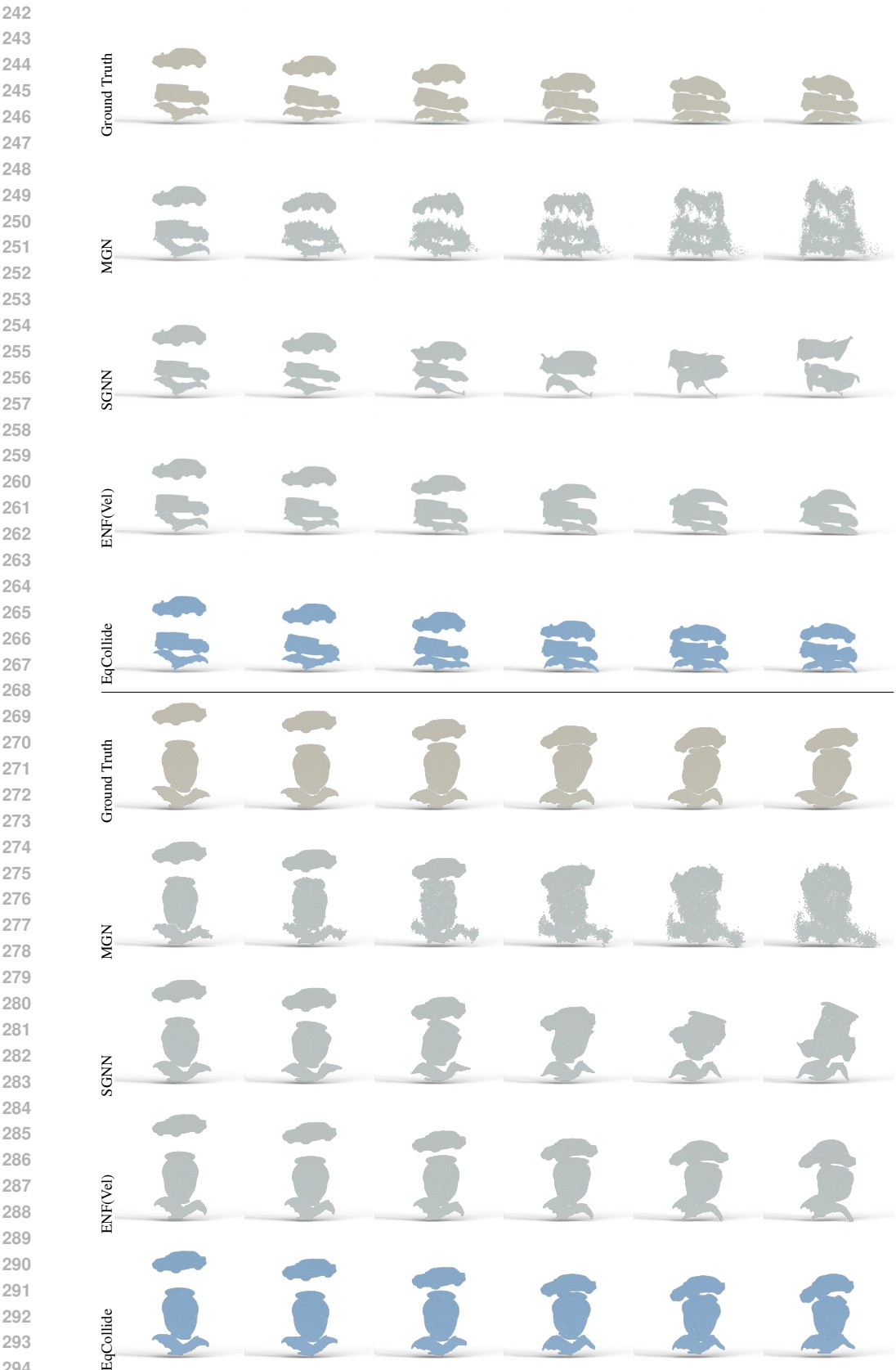

Figure 15: Visualization for rollout prediction results of EqCollide and baselines on two testing samples with unseen 3 objects.

Table 7: MSE of EqCollide after finetuning on four-object collision samples

| Prediction | Prediction error in unseen 4 objects($\times 10^{-6}$) $\downarrow$ | | | | | |
|---|---|---|---|---|---|---|
| scenario | 1-step | 5-step | 10-step | 15-step | 20-step | 25-step |
| EqCollide | 0.003 | 2.095 | 9.148 | 22.971 | 49.821 | 109.918 |

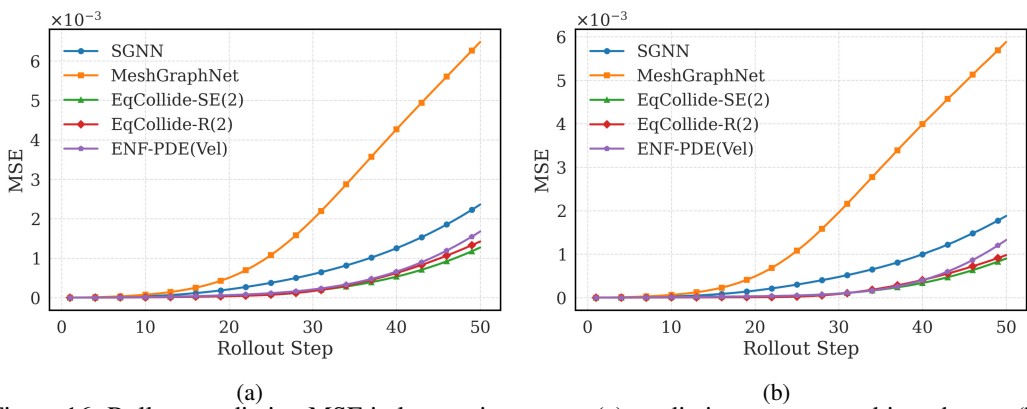

(a)                                                           (b)

Figure 16: Rollout prediction MSE in longer time steps: (a) prediction on unseen object shapes, (b) prediction on unseen object combinations.

## D.4 GENERALIZATION ABILITY TO LONGER ROLLOUT STEPS

We compare the MSE of predicted positions between EqCollide and baseline models over extended rollout steps. As illustrated in Figure 16, across both test sets, the predicted displacements from EqCollide exhibit significantly delayed divergence compared to the baselines as the number of rollout steps increases. This result demonstrates the robustness of EqCollide in maintaining accurate predictions over longer temporal horizons.

It is worth noting that, although both our visualizations and tabulated results in the main paper show that the $\mathbb{R}^n$-equivariant version of EqCollide outperforms its $\mathbf{SE}(n)$ counterpart, this trend reverses as we further extend the rollout time horizon. The $\mathbf{SE}(n)$-equivariant model begins to surpass the $\mathbb{R}^n$-equivariant model at longer extrapolation steps. The higher long-term rollout accuracy of SE(n) model can be attributed to its stronger constraints are symmetry. Non-equivariant models or models with fewer constraints can introduce small symmetry-violating errors at each step. These errors tend to compound over long rollouts, especially in dynamical systems. In contrast, the SE(n) model ensures that the learned physics remains consistent regardless of the object's orientation or position, leading to significantly more stable and physically plausible rollouts over long horizons. This leads to more stable long-term trajectories.

## D.5 ROBUSTNESS TO SPATIAL SAMPLING DENSITY

In the 2D collision scenario, we randomly sampled the initial positions and velocities of 1000, 2000, 4000, and 8000 points from the ground truth and fed them into EqCollide for rollout prediction. The predicted trajectories were then compared against the ground truth. As shown in Table 8, the displacement prediction error changes only marginally as the spatial resolution varies across different rollout steps. Importantly, this trend is consistent across both testing sets. It confirms that EqCollide is robust to variations in spatial sampling density during inference.

Table 8: Displacement prediction MSE under different spatial sampling density

| Number of | MSE in unseen object combination($\times 10^{-6}$) $\downarrow$ | | | | | | MSE in unseen object shape($\times 10^{-6}$) $\downarrow$ | | | | | |
|---|---|---|---|---|---|---|---|---|---|---|---|---|
| sampling points | 1-step | 5-step | 10-step | 15-step | 20-step | 25-step | 1-step | 5-step | 10-step | 15-step | 20-step | 25-step |
| 1000 | 0.0020 | 1.1938 | 4.4648 | 7.4907 | 10.5033 | 30.4332 | 0.0013 | 1.7533 | 8.0106 | 18.8392 | 36.6904 | 79.1584 |
| 2000 | 0.0020 | 1.2074 | 4.3471 | 7.1669 | 10.1806 | 29.5102 | 0.0012 | 1.8299 | 8.6910 | 20.7233 | 39.8498 | 82.3349 |
| 4000 | 0.0024 | 1.2520 | 4.4316 | 7.8950 | 11.5292 | 31.3461 | 0.0014 | 1.8378 | 8.6648 | 20.9657 | 40.5153 | 83.1981 |
| 8000 | 0.0024 | 1.2466 | 4.5920 | 7.7619 | 11.1021 | 30.7984 | 0.0015 | 1.8543 | 8.8773 | 21.5990 | 42.5936 | 87.1033 |

## D.6 SENSITIVITY TO COLLISION-DETECTION HYPERPARAMETERS

We quantitatively analyze the influence of collision threshold and collision region radius on the performance of EqCollide. In one experiment, we set the collision threshold to 0.03, 0.05 (our default), and 0.07. In the other experiment, we set the collision radius to 0.03, 0.05 (our default), and 0.07. After model training for 800 epochs, the rollout prediction results are generated and summarized in Table 9. The results show that using both the collision threshold and collision detection radius of 0.05, which is our default setting, yields the best overall rollout accuracy. The model trained with a threshold of 0.07 and a radius of 0.05 achieves the lowest error in the first 5 rollout steps, but incurs higher error when the rollout horizon exceeds 5 steps. Nevertheless, the performance does not degrade significantly under different hyperparameter settings. This suggests that EqCollide is reasonably robust to the choice of collision threshold and radius within a sensible range.

Table 9: Displacement prediction MSE under different collision-detection hyperparameters

| Collision-detection | MSE in unseen object combination($\times 10^{-6}$) ↓ | | | | | | MSE in unseen object shape($\times 10^{-6}$) ↓ | | | | | |
| hyperparameters | 1-step | 5-step | 10-step | 15-step | 20-step | 25-step | 1-step | 5-step | 10-step | 15-step | 20-step | 25-step |
| --- | --- | --- | --- | --- | --- | --- | --- | --- | --- | --- | --- | --- |
| threshold=0.05; radius=0.05 | 0.0042 | 2.0544 | **4.9485** | **7.7465** | **12.0162** | **28.6257** | 0.0025 | 2.0123 | **7.3596** | **17.6444** | **34.0980** | **63.9872** |
| threshold=0.05; radius=0.03 | 0.0033 | 1.9812 | 9.8418 | 22.4091 | 38.4926 | 76.4884 | 0.0023 | 1.9763 | 11.1196 | 30.0354 | 57.8808 | 109.3291 |
| threshold=0.05; radius=0.07 | 0.0034 | 1.8174 | 6.9708 | 14.1883 | 24.6769 | 56.1725 | 0.0022 | 1.9993 | 8.5377 | 20.0746 | 40.6281 | 84.8553 |
| threshold=0.03; radius=0.05 | 0.0045 | 2.0584 | 7.8874 | 16.4094 | 26.7681 | 56.1529 | 0.0036 | 2.1808 | 10.4197 | 26.1436 | 48.6647 | 94.8254 |
| threshold=0.07; radius=0.05 | **0.0031** | **1.7482** | 5.6350 | 9.7891 | 15.4129 | 38.6236 | **0.0021** | **1.8351** | 8.3419 | 20.5063 | 38.7146 | 78.0397 |

*Note: Threshold denotes collision threshold and radius denotes collision region radius.

## D.7 ANALYSIS ON COMPUTATIONAL EFFICIENCY

To provide a clearer picture of computational efficiency, we include a quantitative comparison of inference time for the 3D cubic collision scenario on a horizontal plane. Specifically, we evaluate all methods on 50 testing samples over 30 time steps. We report the average inference time per sample per time step. All experiments are run under identical hardware settings.

Table 10: Inference time comparison for different models.

| Model | Average inference time (s) |
| --- | --- |
| EqCollide | 0.058709 |
| SGNN | 0.330947 |
| MeshGraphNets | 0.563553 |
| ENF-PDE | 0.044245 |

The results are summarized in Table 10, EqCollide achieves significantly lower computational cost than most competing approaches, with only a slightly higher runtime than ENF-PDE. ENF-PDE is the fastest method because it also operates on a compact control-point graph and does not perform collision detection, which reduces overhead. EqCollide, in contrast, incorporates collision-aware message passing and end-to-end equivariance, which adds some computation but brings additional benefits.

## THE USE OF LARGE LANGUAGE MODELS

In this work, large language models (LLMs) were used solely for language polishing and grammar checking.

