# OpenReview forum: "EqCollide: Equivariant and Collision-Aware Deformable Objects Neural Simulator"
_ICLR.cc/2026/Conference — Submitted to ICLR 2026_

### Official Review · Reviewer_k92Y · 2025-10-26

**Soundness:** 3
**Presentation:** 3
**Contribution:** 3
**Rating:** 4
**Confidence:** 4

**Summary:**

This paper introduces EqCollide, a novel simulator for deformable object collisions. The framework's core contribution is an end-to-end $SE(n)$ equivariant architecture, which is composed of three equivariant components: a GNN-based NODE processor to simulate the latent dynamics, and an equivariant neural field encoder and decoder to extract and reconstruct the continuous velocity field.
Additionally, the paper proposes a collision-aware message-passing mechanism that operates on this compact latent control point graph.
Experimental results show that EqCollide achieves lower MSE and more stable long-term rollouts compared to existing (including semi-equivariant) baselines in both 2D and 3D collision scenarios.

**Strengths:**

The paper's most significant strength is its strict, full-pipeline equivariant design. Ensuring that every component from the encoder to the decoder preserves equivariance is theoretically more robust than non-equivariant or semi-equivariant approaches (like GNS or ENF-PDE).
The paper demonstrates convincingly low error in its primary 2D and 3D scenarios. The experiments in Figure 3 and Figure 5 effectively demonstrate the model's robustness to out-of-distribution geometric transformations (translations, rotations), which is a key failure point for non-equivariant models.

**Weaknesses:**

1.The 3D experiments are the paper's primary weakness. All 3D results appear to be limited to cubes or cube-like soft bodies with minor cutouts. This makes it impossible to assess the model's ability to handle complex geometries. Even experiments with simple 3D objects, such as dominoes, or standard graphics models like the "bunny" are missing.

2.The paper only demonstrates fine-tuning from 2-object to 3-object collisions (Figure 6). This is a very preliminary test of scalability. A truly useful simulator should be able to handle complex interactions between many (e.g., 10+) objects. It is unclear if the proposed simulator remains stable and efficient as the number of control points increases significantly.


3.The paper shows that SGNN performs poorly but does not deeply analyze the root cause of its failure. The authors should clarify whether SGNN's high error is due to the discreteness of its spatio-temporal representation, its lack of end-to-end equivariant design, or its mechanism of rebuilding the graph at each frame. This analysis is critical for highlighting the specific advantages of EqCollide's components (e.g., the continuous neural field or the equivariance). Finally, it would be nice to see some examples showing spatial or temporal "resolution invariance" in EqCollide.

**Questions:**

As I mentioned before, the authors should strengthen the paper by providing more extensive experiments on complex 3D shapes and multi-object scalability, while also clarifying the specific reasons for baseline failures.

---

> ### Author Response · Authors · 2025-11-25
> **Official Response to Reviewer k92Y (1)**
>
> We sincerely thank the reviewer for the constructive feedback. We are grateful for your positive assessment of our work's soundness, presentation, and contribution, specifically regarding the strict full-pipeline equivariant design and the model's robustness to out-of-distribution geometric transformations. Your comments have helped us identify areas to improve the breadth of our experiments and the depth of our baseline analysis. We have revised the manuscript accordingly.
>
> ---
> ### Responses to Weaknesses
>
> > 1. The  3D experiments are the paper's primary weakness. All 3D results appear  to be limited to cubes or cube-like soft bodies with minor cutouts. This  makes it impossible to assess the model's ability to handle complex  geometries. Even experiments with simple 3D objects, such as dominoes,  or standard graphics models like the "bunny" are missing.
>
> Thank you for highlighting the importance of evaluating on more complex 3D geometries. We agree that testing beyond simple shapes is crucial for demonstrating real-world applicability. Following your suggestion, we have added a new experiment involving two 3D cow-shaped soft bodies undergoing collision, which represents a much more complex geometry than the cube-like structures in the original manuscript.
>
> Specifically, we simulate a scenario where two elastic cows collide under different initial velocities and incident angles. We evaluate our method against the three baselines reported in the original paper. **The displacement prediction results of all models are summarized in the table below.** These results, together with the rollout visualizations in Figure 6 of the revised manuscript, demonstrate that EqCollide effectively learns collision dynamics even on such complex geometries. EqCollide achieves lower displacement prediction error compared to all baseline models when the rollout step is larger than 5. Notably, while SGNN performs better in the initial 5 time steps, it demonstrates poor capability in long-term rollouts.
>
> | Displacement prediction MSE | 1-step MSE (×10⁻⁶)  | 5-step MSE (×10⁻⁶)  | 10-step MSE (×10⁻⁶) | 15-step MSE (×10⁻⁶)  | 20-step  MSE (×10⁻⁶) | 25-step MSE (×10⁻⁶)  |
> |---------------------|---------|---------|---------|---------|---------|---------|
> | ENF-PDE (Vel)       | 0.0071   | 0.1125   | 0.5881   | 1.7226   | 3.8218   | 7.2454   |
> | MeshGraphNets       | 0.0414   | 0.4874   | 1.9395   | 4.8680   | 10.0842  | 18.7400  |
> | SGNN                | **0.0008** | **0.0513** | 0.5133   | 2.0614   | 5.5260   | 11.8415  |
> | EqCollide               | 0.0034   | 0.0931   | **0.4873** | **1.3516** | **2.8159** | **4.9163** |
>
> All of these results have been incorporated into Section 4.3 of the revised manuscript. The corresponding collision videos have also been added to the anonymous Git repository, and we kindly invite the reviewer to view them via the provided in [anonymous Git repository](https://anonymous.4open.science/r/EqCollide-F3E9/) in Reproducibility Statement.
>
> ---

---

> ### Author Response · Authors · 2025-11-25
> **Official Response to Reviewer k92Y (2)**
>
> > 2. The paper only demonstrates fine-tuning from 2-object to 3-object  collisions (Figure 6). This is a very preliminary test of scalability. A  truly useful simulator should be able to handle complex interactions  between many (e.g., 10+) objects. It is unclear if the proposed  simulator remains stable and efficient as the number of control points  increases significantly.
>
> Thank you for this insightful suggestions. In the revised manuscript, we further **increase the number of interacting objects to four and finetune EqCollide** on these samples. The visualization of the rollout predictions in Figure 14 of the revised manuscript shows that our model **still produces visually plausible and physically coherent collision dynamics**, confirming its ability to generalize to more complex multi-body scenarios. The displacement prediction MSE at different rollout steps is reported in Table 7 (as below) in the revised manuscript. As shown in the table, the majority of the errors remain within the same order of magnitude as the error in three-object collisions. Importantly, this level of deviation does not compromise the physical plausibility of the predicted motion, as also reflected in the rollout visualizations in Figure 14.
>
>
> | Displacement prediction MSE after finetuning| 1-step MSE (×10⁻⁶)  | 5-step MSE (×10⁻⁶)  | 10-step MSE (×10⁻⁶) | 15-step MSE (×10⁻⁶)  | 20-step  MSE (×10⁻⁶) | 25-step MSE (×10⁻⁶)  |
> |---------------------|---------|---------|---------|---------|---------|---------|
> | EqCollide               | 0.003   | 2.095   | 9.148 | 22.971 | 49.821 | 109.918 |
>
> Considering that increasing the number of objects beyond four is unlikely to introduce fundamentally new interaction patterns beyond those already captured in the four-object case, and given the time limitations as well as the bounded size of our simulation domain, we have not further extended the experiments. A more systematic study of highly crowded multi-object collisions would be an interesting extension, and we plan to explore this direction in future work.
>
> ---

---

> ### Author Response · Authors · 2025-11-25
> **Official Response to Reviewer k92Y (3)**
>
> > 3. The paper shows that SGNN performs poorly but does not deeply  analyze the root cause of its failure. The authors should clarify whether SGNN's high error is due to the discreteness of its  spatio-temporal representation, its lack of end-to-end equivariant  design, or its mechanism of rebuilding the graph at each frame. This  analysis is critical for highlighting the specific advantages of  EqCollide's components (e.g., the continuous neural field or the  equivariance). Finally, it would be nice to see some examples showing  spatial or temporal "resolution invariance" in EqCollide.
>
> We agree that a deeper analysis of the SGNN baseline is beneficial to highlight the advantages of EqCollide. We've identified two main causes of SGNN's degraded performance:
>
> - **Lack of spatially continuous representation**: SGNN relies on a purely discrete spatiotemporal graph. Without a continuous neural representation (as provided by our conditional neural field), it struggles to maintain deformation consistency during later collision stages. This leads to shape divergence, as illustrated in Figure 4, Figure 6, Figure 10, Figure 11, Figure 12, and Figure 13.
>
> - **Inefficient message passing due to large graph size**: SGNN performs message passing over all mass points, resulting in a large graph. In contrast, EqCollide operates on a compact set of latent control points, which significantly improves information propagation efficiency.
>
> Equivariance might not be the reason for SGNN failure. Because SGNN is equivariant to translation and rotation around gravity, which has been introduced in [1]. The clarifications and discussion above have been added into Section 4.3 in the revised manuscript.
>
> To further strengthen our argument, we have conducted an additional experiment to evaluate EqCollide's **spatial resolution invariance**. In the 2D collision scenario, during inference, we randomly sampled the initial positions and velocities of **1000, 2000, 4000, and 8000 points** from the ground truth and fed them into EqCollide for rollout prediction. The predicted trajectories were then compared against the ground truth. The experiment has been added to **Appendix D.5 and summarized in Table 8 (as follows) in the updated manuscript**. As shown in Table 8, the displacement prediction error changes only marginally as the spatial resolution varies across different rollout steps and test sets. These results confirm that EqCollide is robust to variations in spatial sampling density during inference.
>
> | Number of sampling points | *MSE-Combo (×10⁻⁶) | *MSE-Combo (×10⁻⁶) | *MSE-Combo (×10⁻⁶) | *MSE-Combo (×10⁻⁶) | *MSE-Combo (×10⁻⁶) | *MSE-Combo (×10⁻⁶) | *MSE-Shape (×10⁻⁶) | *MSE-Shape (×10⁻⁶) | *MSE-Shape (×10⁻⁶) | *MSE-Shape (×10⁻⁶) | *MSE-Shape (×10⁻⁶) | *MSE-Shape (×10⁻⁶) |
> |---------------------------|----------------------|-----|--------|--------|--------|---------|-----------------------|-----|--------|--------|--------|---------|
> |                           | 1-step | 5-step | 10-step | 15-step | 20-step | 25-step | 1-step | 5-step | 10-step | 15-step | 20-step | 25-step |
> | 1000                      | 0.0020  | 1.1938 | 4.4648 | 7.4907 | 10.5033 | 30.4332 | 0.0013 | 1.7533 | 8.0106 | 18.8392 | 36.6904 | 79.1584 |
> | 2000                      | 0.0020  | 1.2074 | 4.3471 | 7.1669 | 10.1806 | 29.5102 | 0.0012 | 1.8299 | 8.6910 | 20.7233 | 39.8498 | 82.3349 |
> | 4000                      | 0.0024 | 1.2520 | 4.4316 | 7.8950 | 11.5292 | 31.3461 | 0.0014 | 1.8378 | 8.6648 | 20.9657 | 40.5153 | 83.1981 |
> | 8000                      | 0.0024 | 1.2466 | 4.5920 | 7.7619 | 11.1021 | 30.7984 | 0.0015 | 1.8543 | 8.8773 | 21.5990 | 42.5936 | 87.1033 |
>
> **Reference**
>
> [1] Han, Jiaqi, et al. "Learning physical dynamics with subequivariant graph neural networks." NeurIPS (2022)
>
> ---
>
> We sincerely thank the reviewer once again for the constructive feedback. Your comments helped us significantly improve the manuscript by broadening our experimental evaluation (including new experiments on complex cow-shaped 3D collisions and spatial resolution invariance), deepening the analysis of baseline limitations (especially SGNN), and clarifying the scalability and design intent of the proposed framework. We have revised the manuscript accordingly and believe the additional results and discussions further strengthen our contribution to collision-aware deformable simulation.

---

### Official Review · Reviewer_LqEa · 2025-10-30

**Soundness:** 2
**Presentation:** 3
**Contribution:** 2
**Rating:** 4
**Confidence:** 2

**Summary:**

This paper proposes EqCollide, a simulator for deformable-object collisions that is end-to-end SE(n)-equivariant and pairs a collision-aware message-passing Neural ODE over learned control points with a neural-field decoder to produce continuous, resolution-independent velocity fields, reporting substantially lower rollout MSE than prior learned baselines.

**Strengths:**

1. The proposed translation-equivariant and collision-aware framework performs well on the physics simulation task and can be zero-shot generalized to different object shapes.
2. Empirical robustness to group-transformed inputs is shown and discussed, highlighting geometric consistency as a benefit of the equivariant design.
3. The architectural specifics are documented, promoting reproducibility.

**Weaknesses:**

1. The SE(n) variant's performance degrades a lot, even worse than EqCollide w/o equivariance in some cases. This largely decreases the claimed equivariant property.
2. The reliance on a collision-detection subsystem with fixed hyperparameters means performance could be sensitive to thresholding and region-radius choices.
3. Although the authors claim that EqCollide achieves high computational efficiency, the paper does not report any wall-clock runtimes, throughput (e.g., steps/s, FPS), FLOPs, or parameter counts for EqCollide or baselines.

**Questions:**

1. How would EqCollide perform for objects with different densities, stiffness, and other particle properties?

I strongly suggest the authors to incorporate a video to better demonstrate the simulated physics results.

---

> ### Author Response · Authors · 2025-11-25
> **Official Response to Reviewer LqEa (1)**
>
> We sincerely thank the reviewer for the effort in evaluating our work, and for the constructive comments and questions. We appreciate your positive assessment of our **translation-equivariant and collision-aware framework**, your recognition of its **robustness to group-transformed inputs**, and your acknowledgement that the **architectural specifics are clearly documented**, which promotes reproducibility. Your feedback has helped us clarify the scope, positioning, and strengths of EqCollide, and we have revised the manuscript accordingly.
>
> ---
> ### Responses to Weaknesses
>
> > 1. The SE(n) variant's performance degrades a lot, even worse than  EqCollide w/o equivariance in some cases. This largely decreases the  claimed equivariant property.
>
> We appreciate this important observation. Indeed, enforcing SE(n)-equivariance can make optimization more challenging and may sometimes lead to worse performance than a less constrained variant, especially under limited data or model capacity. Similar conclusions have been made in other equivariant works, such as  ENF [1], where $R^n$ equivariance yielded better accuracy than SE(n), and in Vector Neuron [2], which found the reconstruction performance of equivariant networks was inferior to their original versions.
>
> Although the SE(n)-equivariant model is intentionally more constrained: it strictly preserves symmetry properties, which can be beneficial for **generalization, physical consistency, and data efficiency**. Whether to impose stronger geometric constraints is **task- and data-dependent**. In regimes with abundant, diverse training data, a non-equivariant or partially equivariant model may achieve similar numerical performance while still implicitly learning some symmetries. In contrast, when data are limited or when strong geometric consistency is particularly important, an explicitly equivariant architecture can offer clear advantages. For instance, AlphaFold2 [3] adopts equivariant attention architecture for enforcing geometric consistency under rigid transformations. AlphaFold3 [4] no longer explicitly uses equivariant structures, instead leveraging large-scale data to implicitly learn spatial relationships
>
> In this paper, we aim to **systematically study an end-to-end equivariant simulator** for deformable collisions and demonstrate its geometric robustness (e.g., under group-transformed inputs) and its ability to generate continuous, resolution-independent velocity fields. Even though the fully SE(n)-equivariant variant may not always beat a non-equivariant baseline on every metric, its **geometric guarantees, better physical interpretability, and robustness to transformations** remain central and meaningful contributions.
>
> We have revised the manuscript in Section 4.2 to explicitly discuss this trade-off and to clarify that our focus is on the **equivariant design and its geometric properties**, rather than solely on the strongest possible numerical performance in every setting.
>
> **Reference**
>
> [1] Wessels, David R., et al. "Grounding continuous representations in geometry: Equivariant neural fields." ICLR (2025)
>
> [2] Deng, Congyue, et al. "Vector neurons: A general framework for so (3)-equivariant networks." ICCV (2021)
>
> [3] Jumper, John, et al. "Highly accurate protein structure prediction with AlphaFold." Nature (2021)
>
> [4]Abramson, Josh, et al. "Accurate structure prediction of biomolecular interactions with AlphaFold 3." Nature (2024)
>
> ---

---

> ### Author Response · Authors · 2025-11-25
> **Official Response to Reviewer LqEa (2)**
>
> > 2. The reliance on a collision-detection subsystem with fixed  hyperparameters means performance could be sensitive to thresholding and  region-radius choices.
>
> Thank you for pointing this out. We agree that the choice of collision-detection hyperparameters could influence performance. We've **added additional experiments** in the revised manuscript to quantitatively analyze the influence of collision thresholds and collision region radius on the performance of EqCollide.
>
> In one experiment, we set the collision thresholds to 0.03, 0.05 (our default), and 0.07. In the other experiment, we set the collision radius to 0.03, 0.05 (our default), and 0.07. After model training, the rollout prediction results are summarized in Table 9 (shown as follows) in Appendix D.6 of revised manuscript. The results show that using both the collision threshold and collision detection radius of 0.05, which is our default setting, yields the best overall rollout accuracy. The model trained with a threshold of 0.07 and a radius of 0.05 achieves the lowest error in the first 5 rollout steps, but incurs higher error when the rollout horizon exceeds 5 steps. Nevertheless, the performance does not degrade significantly under different hyperparameter settings. This suggests that EqCollide is reasonably robust to the choice of collision threshold and radius within a sensible range.
>
> | Collision-detection hyperparameters | \*MSE-Combo (×10⁻⁶) | \*MSE-Combo (×10⁻⁶) | \*MSE-Combo (×10⁻⁶) | \*MSE-Combo (×10⁻⁶) | \*MSE-Combo (×10⁻⁶) | \*MSE-Combo (×10⁻⁶) | \*MSE-Shape (×10⁻⁶) | \*MSE-Shape (×10⁻⁶) | \*MSE-Shape (×10⁻⁶) | \*MSE-Shape (×10⁻⁶) | \*MSE-Shape (×10⁻⁶) | \*MSE-Shape (×10⁻⁶) |
> |------------------------------------|------------------------|-----|--------|--------|--------|---------|------------------------|-----|--------|--------|--------|---------|
> |                                    | 1-step | 5-step | 10-step | 15-step | 20-step | 25-step | 1-step | 5-step | 10-step | 15-step | 20-step | 25-step |
> | threshold=0.05; radius=0.05        | 0.0042 | 2.0544 | **4.9485** | **7.7465** | **12.0162** | **28.6257** | 0.0025 | 2.0123 | **7.3596** | **17.6444** | **34.0980** | **63.9872** |
> | threshold=0.05; radius=0.03        | 0.0033 | 1.9812 | 9.8418 | 22.4091 | 38.4926 | 76.4884 | 0.0023 | 1.9763 | 11.1196 | 30.0354 | 57.8808 | 109.3291 |
> | threshold=0.05; radius=0.07        | 0.0034 | 1.8174 | 6.9708 | 14.1883 | 24.6769 | 56.1725 | 0.0022 | 1.9993 | 8.5377 | 20.0746 | 40.6281 | 84.8553 |
> | threshold=0.03; radius=0.05        | 0.0045 | 2.0584 | 7.8874 | 16.4094 | 26.7681 | 56.1529 | 0.0036 | 2.1808 | 10.4197 | 26.1436 | 48.6647 | 94.8254 |
> | threshold=0.07; radius=0.05        | **0.0031** | **1.7482** | 5.6350 | 9.7891 | 15.4129 | 38.6236 | **0.0021** | **1.8351** | 8.3419 | 20.5063 | 38.7146 | 78.0397 |
>
> ---
>
> > 3. Although the authors claim that EqCollide achieves high  computational efficiency, the paper does not report any wall-clock  runtimes, throughput (e.g., steps/s, FPS), FLOPs, or parameter counts for EqCollide or baselines.
>
> We appreciate this suggestion and have added quantitative runtime comparisons to make our efficiency claims more concrete.
>
> To provide a clearer picture of computational efficiency, we now include a **quantitative comparison of inference time** for the 3D cubic collision scenario on a horizontal plane. Specifically:
>
> - We evaluate all methods on **50 testing samples** over **30 time steps**.
> - We report the **average inference time per sample per time step**.
> - All experiments are run under identical hardware settings (**H800 GPU**).
>
> The results are summarized in the following table (which has been added as **Table 10** in the revised manuscript):
>
> | Model        | Average One-Step Inference Time (s) |
> |-------------|--------------------------------------|
> | EqCollide    | 0.058709                             |
> | SGNN         | 0.330947                             |
> | MeshGraphNets | 0.563553                             |
> | ENF-PDE      | 0.044245                             |
>
> As shown in the table, **EqCollide achieves significantly lower computational cost than most competing approaches**, with only a slightly higher runtime than ENF-PDE:
>
> - **ENF-PDE** is the fastest method because it also operates on a **compact control-point graph** and **does not perform collision detection**, which reduces overhead.
> - **EqCollide**, in contrast, incorporates **collision-aware message passing and end-to-end equivariance**, which adds negligible computation time but brings additional benefits.
>
> We emphasize that **EqCollide provides stronger geometric consistency and clearer physical interpretability**, thanks to its end-to-end equivariant design and explicit collision-aware architecture. These advantages, combined with its competitive runtime, make EqCollide a compelling choice for robust and generalizable deformable collision modeling.
>
> ---

---

> ### Author Response · Authors · 2025-11-25
> **Official Response to Reviewer LqEa (3)**
>
> ### Responses to Question
>
> > 1. How would EqCollide perform for objects with different densities, stiffness, and other particle properties?
>
> We appreciate this insightful question. In our original experiments, we have already evaluated the performance of EqCollide on datasets with **different material settings**. Specifically:
>
> - **Poisson's ratio:** fixed at **0.4** in both 2D and 3D scenarios.
> - **2D simulations:** Young's modulus is set to **2000 Pa**.
> - **3D cubic block and 3D alphabet block collision:** Young's modulus is set to **300 Pa**.
> - **3D cow collision:** Young's modulus is set to **200 Pa**.
>
> Although these experiments do not yet span a wide range of densities and stiffnesses, they demonstrate that EqCollide can successfully learn the dynamics of deformable objects under different elastic regimes.
>
> However, **generalization across diverse material properties** is another story. Although it is an important but also challenging direction for physical simulation, it is **not the primary focus** of this manuscript. In this work, our **main contribution** is to propose an **end-to-end equivariant and collision-aware framework** for deformable-body collisions, using learned control points and a neural-field decoder. We focus on demonstrating strong generalization across  **different object shape combinations**, **unseen geometries** at test time, and scaling from two-object to multi-object interactions. These experiments show that EqCollide can robustly model complex collision dynamics under substantial variation in **geometry and interaction patterns**, which we believe is highly non-trivial and central to the paper's goals.
>
> We believe that generalizing simulators across **multiple material regimes** is a valuable direction, and we plan to investigate this in future work. In the revised manuscript, we have clarified the material settings used in our current experiments in **Appendix C.1** and explicitly highlighted **multi-material generalization as future work** in the conclusion.
>
> ---
>
> ### Responses to Suggestion
>
> > 1. On video demonstrations of simulated physics results
>
> As indicated in the original submission, we have already included **rollout videos** of our simulation results in the [anonymous Git repository](https://anonymous.4open.science/r/EqCollide-F3E9/) mentioned in the **REPRODUCIBILITY STATEMENT**. We have also updated these videos to include the **new 3D cow-shaped deformable-body collision** experiments. We kindly invite the reviewer to view them via that link.
>
> ---
>
> We sincerely thank the reviewer again for the constructive feedback and for recognizing the strengths of our equivariant, collision-aware framework. We hope that the additional experiments, clarifications, and discussions adequately address your concerns and help convey the contribution of EqCollide more clearly.

---

### Official Review · Reviewer_VqQ7 · 2025-10-31

**Soundness:** 3
**Presentation:** 3
**Contribution:** 3
**Rating:** 6
**Confidence:** 4

**Summary:**

EqCollide is introduced as an end-to-end equivariant simulator for collisions of deformable bodies. It adopts an Encoder-Processor-Decoder architecture: an equivariant variant of PointNet++ encodes high-resolution mass points into a small set of control points; a collision-aware equivariant GNN-NODE models the dynamics on those control points; finally, an equivariant neural field reconstructs a continuous velocity field for rollout prediction. The authors build a new dataset, DeformableObjectsCollision (2D/3D, three scenario types). Compared with baseline methods, EqCollide attains better or comparable displacement MSE and visual quality in both 2D and 3D, and demonstrates robustness to coordinate-frame transformations and scalability.

**Strengths:**

1.	Equivariance is maintained end-to-end-from encoding to decoding-avoiding the common gap where inputs/outputs are equivariant but the latent state is not; the new decoder is carefully designed for vector-field covariance.
2.	Cross-object message passing is triggered only when local collisions are detected, which is both efficient and physically intuitive for contact-response; the threshold and radius are explicitly tunable.
3.	A small number of control points drives a continuous velocity field, balancing expressivity and efficiency and reducing sensitivity to resolution.
4.	The proposed dataset covers three types of 2D/3D collisions. EqCollide outperforms strong baselines in long-horizon rollouts and under coordinate changes, with ablations validating the roles of equivariance and collision awareness.

**Weaknesses:**

1.	As mentioned by the authors, stronger SE(n) equivariance constraints increase optimization complexity; on short-horizon tests, the SE(n) variant does not surpass the R^n version or the strongest baseline-reflecting a trade-off between geometric consistency and average accuracy. It would help to report training stability/hyperparameter sensitivity and to explain mechanistically why SE(n) performs better over long horizons.
2.	In three-object scenarios, additional finetuning is required to reach desirable accuracy. While zero-shot results show reasonable trends, the errors remain large, suggesting limited plug-play generalization to more complex multi object interactions.
3.	Throughout the experiments, the collision threshold and local radius are fixed at 0.05. It is unclear whether such a preset is robust across different scales, materials, and velocity distributions.

**Questions:**

see above

---

> ### Author Response · Authors · 2025-11-25
> **Official Response to Reviewer VqQ7 (1)**
>
> We sincerely thank the reviewer for the positive assessment and the constructive feedback. We appreciate your recognition of EqCollide's strengths. Below, we address your specific questions regarding optimization trade-offs, generalization, and hyperparameter robustness.
>
> ---
> ### Responses to Weaknesses
>
> > 1. As mentioned by the authors, stronger SE(n) equivariance  constraints increase optimization complexity; on short-horizon tests, the SE(n) variant does not surpass the R^n version or the strongest  baseline-reflecting a trade-off between geometric consistency and  average accuracy. It would help to report training  stability/hyperparameter sensitivity and to explain mechanistically why SE(n) performs better over long horizons.
>
> We agree with the reviewer that enforcing strict $SE(n)$ equivariance introduces a stronger inductive bias. This restricts the optimization landscape, potentially making short-term fitting ("training accuracy") more challenging and increasing the prediction error compared to unconstrained $R^n$ models. Similar conclusions have been made in other equivariant works, such as ENF [1], where $R^n$ equivariance yielded better accuracy than SE(n), and in Vector Neuron [2], which found that the reconstruction performance of equivariant networks was inferior to their original versions.
>
> As for the training stability, we did encounter training instability with SE(n), characterized by sporadic loss explosions when a constant learning rate was used in our experiments. We addressed this by implementing a cosine learning rate decay schedule, which effectively stabilized the training process and virtually eliminated these anomalies.
>
> The higher long-term accuracy of SE(n) model can be attributed to its stronger constraints in symmetry. Non-equivariant models or models with fewer constraints can introduce small symmetry-violating errors at each step. These errors tend to **compound over long rollouts**, especially in dynamical systems. In contrast, the SE(n) model ensures that the learned physics remains consistent regardless of the object's orientation or position, leading to significantly more stable and physically plausible rollouts over long horizons. This leads to **more stable long-term trajectories**. We've expanded this explanation in Appendix D.4 in the revised version to make the long-horizon benefit of SE(n) equivariance more explicit.
>
> **Reference**
>
> [1] Wessels, David R., et al. "Grounding continuous representations in geometry: Equivariant neural fields." ICLR (2025)
>
> [2] Deng, Congyue, et al. "Vector neurons: A general framework for so (3)-equivariant networks." ICCV (2021)
>
> ---

---

> ### Author Response · Authors · 2025-11-25
> **Official Response to Reviewer VqQ7 (2)**
>
> > 2.  In three-object scenarios, additional finetuning is required to reach desirable accuracy. While zero-shot results show reasonable  trends, the errors remain large, suggesting limited plug-play  generalization to more complex multi object interactions.
>
> We agree that the need for finetuning influence the plug-and-play generalization capacity of the model. However, we would like to emphasize that **achieving fully zero-shot generalization from two-object training to three-object interactions is intrinsically difficult.** Because the dynamics of multi-body interactions involve complex phenomena, such as occlusion and multi-body force chains, that are not fully captured by pairwise training samples alone. Regarding our method, we'd like to emphasize two key observations:
>
> - **Physically consistent behavior:** While zero-shot error is higher, EqCollide still captures the correct physical trends. It suggests that the learned dynamics are applicable beyond the training regime.
>
> - **Efficient Adaptation:** The model achieves desirable accuracy with only **minimal finetuning**. This suggests that the learned representation is flexible and that the fundamental physics are transferable, even if the emergent complexity of multi-body scenarios requires slight adaptation.
>
> To strengthen this point, we additionally conduct a new experiment by finetuning EqCollide on samples involving **four interacting objects.** The rollout predictions shown in Figure 14 of the revised manuscript remain visually plausible and physically coherent, and the displacement prediction MSE at different rollout steps are summarized in Table as below. As shown in the table, the majority of the errors remain within the same order of magnitude as the error in three-object collisions. Importantly, this level of deviation does not compromise the physical plausibility of the predicted motion, as also reflected in the rollout visualizations in Figure 14.
>
> | Displacement prediction MSE after finetuning| 1-step MSE (×10⁻⁶)  | 5-step MSE (×10⁻⁶)  | 10-step MSE (×10⁻⁶) | 15-step MSE (×10⁻⁶)  | 20-step  MSE (×10⁻⁶) | 25-step MSE (×10⁻⁶)  |
> |---------------------|---------|---------|---------|---------|---------|---------|
> | EqCollide               | 0.003   | 2.095   | 9.148 | 22.971 | 49.821 | 109.918 |
>
> Overall, while full zero-shot generalization remains challenging, the experimental results demonstrate that EqCollide can be effectively adapted to increasingly complex collision scenarios with minimal effort. We view this as a strong foundation, and future work will explore explicit multi-object message passing to further bridge the zero-shot gap.
>
> ---

---

> ### Author Response · Authors · 2025-11-25
> **Official Response to Reviewer VqQ7 (3)**
>
> > 3. Throughout the experiments, the collision threshold and local radius are fixed at 0.05. It is unclear whether such a preset is robust  across different scales, materials, and velocity distributions.
>
> To address this concern, we conducted additional ablation studies to test the model's sensitivity to the collision threshold and local radius hyperparameters. We fixed one parameter at the baseline value ($0.05$) and swept the other through $\{0.03, 0.05, 0.07\}$.
> The impact on model performance (measured by the 25-step MSE in the 2D EqCollide experiment) is summarized **Table 9 in Appendix D.6 of revised manuscript**, which is also shown as follows. The results show that using both the collision threshold and collision detection radius of 0.05, which is our default setting, yields the best overall rollout accuracy. The model trained with a threshold of 0.07 and a radius of 0.05 achieves the lowest error in the first 5 rollout steps, but incurs higher error when the rollout horizon exceeds 5 steps.
>
> | Collision-detection hyperparameters | \*MSE-Combo (×10⁻⁶) | \*MSE-Combo (×10⁻⁶) | \*MSE-Combo (×10⁻⁶) | \*MSE-Combo (×10⁻⁶) | \*MSE-Combo (×10⁻⁶) | \*MSE-Combo (×10⁻⁶) | \*MSE-Shape (×10⁻⁶) | \*MSE-Shape (×10⁻⁶) | \*MSE-Shape (×10⁻⁶) | \*MSE-Shape (×10⁻⁶) | \*MSE-Shape (×10⁻⁶) | \*MSE-Shape (×10⁻⁶) |
> |------------------------------------|------------------------|-----|--------|--------|--------|---------|------------------------|-----|--------|--------|--------|---------|
> |                                    | 1-step | 5-step | 10-step | 15-step | 20-step | 25-step | 1-step | 5-step | 10-step | 15-step | 20-step | 25-step |
> | threshold=0.05; radius=0.05        | 0.0042 | 2.0544 | **4.9485** | **7.7465** | **12.0162** | **28.6257** | 0.0025 | 2.0123 | **7.3596** | **17.6444** | **34.0980** | **63.9872** |
> | threshold=0.05; radius=0.03        | 0.0033 | 1.9812 | 9.8418 | 22.4091 | 38.4926 | 76.4884 | 0.0023 | 1.9763 | 11.1196 | 30.0354 | 57.8808 | 109.3291 |
> | threshold=0.05; radius=0.07        | 0.0034 | 1.8174 | 6.9708 | 14.1883 | 24.6769 | 56.1725 | 0.0022 | 1.9993 | 8.5377 | 20.0746 | 40.6281 | 84.8553 |
> | threshold=0.03; radius=0.05        | 0.0045 | 2.0584 | 7.8874 | 16.4094 | 26.7681 | 56.1529 | 0.0036 | 2.1808 | 10.4197 | 26.1436 | 48.6647 | 94.8254 |
> | threshold=0.07; radius=0.05        | **0.0031** | **1.7482** | 5.6350 | 9.7891 | 15.4129 | 38.6236 | **0.0021** | **1.8351** | 8.3419 | 20.5063 | 38.7146 | 78.0397 |
>
> Nevertheless, the performance does not degrade significantly under different hyperparameter settings. It indicates that our method is generally **robust to parameter changes within this range**. While $0.05$ serves as a strong empirical baseline for our dataset, these parameters can be slightly adjusted to optimize performance for specific collision scales or material stiffness requirements in different application scenarios without causing model failure.

---

### Official Review · Reviewer_eMaX · 2025-11-04

**Soundness:** 3
**Presentation:** 3
**Contribution:** 3
**Rating:** 6
**Confidence:** 3

**Summary:**

The paper proposes a learnable simulator designed for collision-aware dynamics modeling, which explicitly models collision for deformable objects. The Encoder uses a PointNet to map the input point cloud to N control points in the latent space, using several selected input features to ensure invariance. The Processer uses a collision-aware GNN-based Neural ODE to model the temporal evolution of the latent system states. The Decoder uses an equivariant neural work to reconstruct the velocity field from the latent states. Then, the Euler integration is used to get the point positions from the velocity field. The networks are trained by optimizing the displacement prediction loss and the reconstruction loss. In the first stage, only the reconstruction loss is used to train the encoder and decoder networks. In the second stage, both the loss are used to train all the network components jointly.

To evaluate the proposed method, the paper proposes a new dataset which consists of 3 different collision scenarios for both 2D and 3D deformable objects. The proposed method is compared against multiple baselines from three types of methods.

**Strengths:**

- The paper proposes a novel method to predict deformable physical systems by utilizing equivariant encoder-processor-decoder models.
- Besides the proposed method, the paper also additionally proposes a new dataset for evaluation.

**Weaknesses:**

I have some additional questions listed in the section below.

**Questions:**

* The collision region is defined by a scalar hyperparameter of the collision region radius. Will there be cases that different control points need to have different radius?
* The dataset primarily involves collisions between two objects, and the paper presents some finetuning results for three-object collisions. How would the method perform when multiple objects interact simultaneously? Will it an issue for the graph neural network to capture long-range interactions across multiple objects? Furthermore, if the dataset were to directly include multi-object collisions (instead of finetuning from two-object data), would that significantly increase the difficulty of training?
* The current dataset focuses on deformable bodies. How would the proposed framework perform when modeling rigid-body collision systems?
* Can the method be extended to handle cases with boundary conditions? for example, when an object collides with a fixed boundary.

---

> ### Author Response · Authors · 2025-11-25
> **Official Response to Reviewer eMaX (1)**
>
> We sincerely thank the reviewer for the constructive feedback. We are grateful for your positive assessment of our work, as well as your recognition that (i) the paper proposes a novel equivariant encoder–processor–decoder architecture for deformable physical systems, and (ii) it introduces a new dataset for systematic evaluation. In response to your valuable comments and questions, we have conducted additional experiments and provide our detailed replies below.
>
> ---
> ### Responses to Questions
>
> > 1. The collision region is defined by a scalar hyperparameter of the collision region radius. Will there be cases that different control points need to have different radius?
>
> Thank you for this question. In the current version of EqCollide, we use a **single collision detection radius** shared across all control points. This design choice stems from following reasons:
>
> - The control points are obtained by a PointNet++-based encoder and are distributed relatively uniformly over the object. Under this construction, a shared radius is sufficient to capture local contact patterns and inter-object interactions.
> - In our experiments across all 2D and 3D scenarios, we did not observe qualitative failure modes that could be attributed to the use of a global radius. The collision-aware GNN effectively captures contact information under this setting.
>
> We agree that using an adaptive collision radius per control point is a meaningful extension. However, it is non-trivial to determine which points should be assigned larger or smaller radius, since collisions can occur at arbitrary locations and the shapes of deformable bodies can vary significantly.
>
> Nevertheless, we have added an additional experiment to **quantitatively analyze the influence of the collision-detection radius** on the performance of EqCollide. Specifically, we set the collision radius to 0.03, 0.05 (our default), and 0.07, and retrain EqCollide on the 2D collision dataset for 800 epochs. The rollout prediction results are summarized in Table 9 (as follows) in Appendix D.6. As shown in the table, using a radius of 0.05 yields the best overall rollout accuracy. Experiments with radius of 0.03 and 0.07 both ahieve less error than our default setting whithin 5 rollout steps but larger error when rollout step exceed 5. However, the overall performance of model under different collision radius remains comparable and does not degrade dramatically. This suggests that EqCollide is **reasonably robust** to the choice of collision radius within a sensible range, even without a more sophisticated adaptive-radius mechanism.
>
> | Collision-detection hyperparameters | \*MSE-Combo (×10⁻⁶) | \*MSE-Combo (×10⁻⁶) | \*MSE-Combo (×10⁻⁶) | \*MSE-Combo (×10⁻⁶) | \*MSE-Combo (×10⁻⁶) | \*MSE-Combo (×10⁻⁶) | \*MSE-Shape (×10⁻⁶) | \*MSE-Shape (×10⁻⁶) | \*MSE-Shape (×10⁻⁶) | \*MSE-Shape (×10⁻⁶) | \*MSE-Shape (×10⁻⁶) | \*MSE-Shape (×10⁻⁶) |
> |------------------------------------|------------------------|-----|--------|--------|--------|---------|------------------------|-----|--------|--------|--------|---------|
> |                                    | 1-step | 5-step | 10-step | 15-step | 20-step | 25-step | 1-step | 5-step | 10-step | 15-step | 20-step | 25-step |
> | threshold=0.05; radius=0.05        | 0.0042 | 2.0544 | **4.9485** | **7.7465** | **12.0162** | **28.6257** | 0.0025 | 2.0123 | **7.3596** | **17.6444** | **34.0980** | **63.9872** |
> | threshold=0.05; radius=0.03        | 0.0033 | 1.9812 | 9.8418 | 22.4091 | 38.4926 | 76.4884 | 0.0023 | 1.9763 | 11.1196 | 30.0354 | 57.8808 | 109.3291 |
> | threshold=0.05; radius=0.07        | 0.0034 | 1.8174 | 6.9708 | 14.1883 | 24.6769 | 56.1725 | 0.0022 | 1.9993 | 8.5377 | 20.0746 | 40.6281 | 84.8553 |
> | threshold=0.03; radius=0.05        | 0.0045 | 2.0584 | 7.8874 | 16.4094 | 26.7681 | 56.1529 | 0.0036 | 2.1808 | 10.4197 | 26.1436 | 48.6647 | 94.8254 |
> | threshold=0.07; radius=0.05        | **0.0031** | **1.7482** | 5.6350 | 9.7891 | 15.4129 | 38.6236 | **0.0021** | **1.8351** | 8.3419 | 20.5063 | 38.7146 | 78.0397 |
>
> ---

---

> ### Author Response · Authors · 2025-11-25
> **Official Response to Reviewer eMaX (2)**
>
> > 2. The dataset primarily involves collisions between two objects, and  the paper presents some finetuning results for three-object collisions.  How would the method perform when multiple objects interact  simultaneously? Will it an issue for the graph neural network to capture  long-range interactions across multiple objects? Furthermore, if the  dataset were to directly include multi-object collisions (instead of  finetuning from two-object data), would that significantly increase the  difficulty of training?
>
> Thank you for this insightful question about multi-object interactions and long-range effects. In the revised manuscript, we further **increase the number of interacting objects to four and finetune EqCollide** on these samples. The visualization of the rollout predictions (see Figure 14 in revised manuscript) shows that our model **still produces visually plausible and physically coherent collision dynamics**, confirming its ability to generalize to more complex multi-body scenarios. The displacement prediction MSE at different rollout steps is reported in Table 7 (as below) in Appendix D.3. As shown in the table, the majority of the errors remain within the same order of magnitude as the error in three-object collisions. Importantly, this level of deviation does not compromise the physical plausibility of the predicted motion, as also reflected in the rollout visualizations in Figure 14. Considering that increasing the number of objects beyond four is unlikely to introduce fundamentally new interaction patterns beyond those already captured in the four-object case, and given the time limitations as well as the bounded size of our simulation domain, we have not further extended the experiments. A more systematic study of highly crowded multi-object collisions would be an interesting extension, and we plan to explore this direction in future work.
>
> | Displacement prediction MSE after finetuning| 1-step MSE (×10⁻⁶)  | 5-step MSE (×10⁻⁶)  | 10-step MSE (×10⁻⁶) | 15-step MSE (×10⁻⁶)  | 20-step  MSE (×10⁻⁶) | 25-step MSE (×10⁻⁶)  |
> |---------------------|---------|---------|---------|---------|---------|---------|
> | EqCollide               | 0.003   | 2.095   | 9.148 | 22.971 | 49.821 | 109.918 |
>
> Regarding **long-range interactions**, EqCollide is designed to enable more efficient and effective message passing. Unlike traditional GNN-based simulators such as MeshGraphNets, which operate on densely connected graphs over all mass particles, EqCollide operates on a **latent control-point graph** and uses a collision-aware GNN-based Neural ODE. Long-range effects are captured through:
>
> - multi-step message passing across the control-point graph, and
> - the continuous neural-field decoder, which propagates information over all the control points in each object.
>
> Finally, if one were to train **directly on multi-object collisions from scratch** (rather than starting from two-object data), this would increase the diversity of contact patterns and require substantially more training data for stable convergence. However, conceptually, this does not pose a fundamental limitation of our architecture, and our few-shot three-object results suggest that EqCollide can be effectively extended toward richer multi-object scenarios.
>
> ---
>
> > 3. The current dataset focuses on deformable bodies. How would the proposed framework perform when modeling rigid-body collision systems?
>
> Thank you for this insightful question. We believe that **EqCollide is, in principle, capable of handling rigid-body collisions**, as a rigid body can be viewed as a special case of a deformable body with **extremely high stiffness parameters**.
>
> From a modeling standpoint:
>
> - **Rigid-body collisions are technically simpler** than deformable collisions: one only needs to track the bodies' **translation and rotation**, without resolving complex elastic deformations.
> - EqCollide is explicitly designed to learn **collision-aware deformation dynamics**. Under our formulation, rigid-body interactions can be regarded as a limiting case where deformations vanish.
>
> Given that our work is primarily motivated by and evaluated on **deformable-object collisions**, where existing learned simulators often struggle, we chose to focus our experiments on deformable systems. We therefore consider additional rigid-body experiments **not essential** to demonstrating the main contribution of this paper. Nevertheless, we agree that extending EqCollide to a broader range of rigid-body benchmarks would be an interesting future direction, and we have noted this in the conclusion in revised version.
>
> ---

---

> ### Author Response · Authors · 2025-11-25
> **Official Response to Reviewer eMaX (3)**
>
> > 4. Can the method be extended to handle cases with boundary conditions? for example, when an object collides with a fixed boundary.
>
> We appreciate this question, and we agree that boundary conditions are important in many physical scenarios. While one of our 3D experiments indeed considers collisions in free space (between two cubical blocks with alphabet-shaped cutouts), **boundary conditions are already present in our current experimental setup**:
>
> - In the **2D collision scenario** (Appendix C.1), deformable objects **fall under gravity and collide with a fixed ground plane**. The ground acts as a **static boundary condition**, and EqCollide successfully models both object–object and object–ground collisions.
> - In one of the **3D scenarios**, the objects **slide and collide on a frictionless plane with ground contact**, which again provides a boundary condition in the form of a smooth supporting surface.
>
> These settings demonstrate that EqCollide naturally supports **fixed boundaries** via its collision-aware message-passing design. We have updated the manuscript to clarify the presence and handling of such boundary conditions so that this aspect is not overlooked.
>
> ---
>
> We thank the reviewer again for the constructive feedback and for the positive overall assessment of our work. Moreover, we've added another 3d collision experiment on two 3d cow-shaped bodies to validate EqCollide's simulation ability in complex shapes. Please check them in Section 4.3 in the revised manuscript. We hope that the additional experiments, clarifications, and discussions address your concerns and further strengthen the case for EqCollide.

---

### Author Response · Authors · 2025-11-25
**General Response**

We thank the reviewers for their constructive feedback. We are encouraged by the positive assessment regarding the novelty of our equivariant architecture, the new collision dataset, and the model's geometric robustness. Based on the reviewers' comments, we have conducted additional experiments and revised the manuscript, highlighting modifications in blue. The major improvements are as follows:
- **We add a new experiment on complex 3D geometries (collisions between two cow-shaped soft bodies).** The results for the new experiment in Section 4.3 show that EqCollide effectively captures deformation dynamics and achieves lower long-term prediction error than baselines. Corresponding videos have been added to the anonymous repository. For more details, see responses to reviewer k92Y.
- **We add a new experiment to validate the generalization ability of EqCollide to more objects collision.** We increase the number of interacting objects to four and finetune EqCollide on these samples. The rollout prediction results show that EqCollide still generates visually plausible and physically coherent collision dynamics, demonstrating good generalization to more complex multi-body scenarios. For more details, see responses to reviewers eMaX, VqQ7, and k92Y.
- **We add a quantitative analysis of hyperparameter sensitivity.** We test varying collision radii and thresholds ($0.03-0.07$). The results in Appendix D.6 (Table 9) demonstrate that EqCollide is robust within a reasonable range, with the default setting ($0.05$) yielding the best overall accuracy. For more details, see responses to reviewers eMaX, VqQ7, and LqEa.
- **We validate the spatial resolution invariance to address concerns about robustness.** By varying the number of inference points ($1000-8000$) in 2D settings, as shown in Appendix D.5 (Table 8), we observe that the displacement prediction error remains stable. This confirms our model's robustness to spatial sampling density. For more details, see responses to reviewer k92Y.
- **We add a quantitative runtime comparison for 3D scenarios.** The results in Table 10 show that EqCollide is significantly faster than SGNN and MeshGraphNets, and competitive with ENF-PDE. This confirms our method balances computational efficiency with geometric consistency. For more details, see responses to reviewer LqEa.
- **We add clarifications on theoretical and practical aspects.** We distinguish the trade-off between SE(n)-equivariance and optimization difficulty (see responses to reviewers VqQ7 and LqEa), clarify the scalability regarding objects with different material (see responses to reviewers eMaX and LqEa), and analyze SGNN’s failure modes due to its lack of continuous representation and inefficient message passing (see responses to reviewer k92Y).

Detailed point-by-point responses are provided below.

---

### Author Response · Authors · 2025-12-03
**Summary of the Discussion Period**

Dear Area Chair,

We thank you for handling our submission and for coordinating the review process. Below we provide a concise summary of how we addressed all reviewer concerns, along with the main contributions and significance of our work to the community.

We are encouraged by the reviewers' positive assessment on the novel **end-to-end equivariant architecture** of our proposed simulator, its higher accuracy compared to strong baselines, and the introduction of a **new collision dataset**. Taken together, we believe EqCollide offers a **substantive advance** in both methodology and benchmarking, and can serve as a **useful building block** for the community’s ongoing efforts in neural simulation and geometric deep learning. We believe our work represents a significant contribution to the ICLR community.

Due to the new ICLR 2026 policy, reviewers were not able to participate in further discussion. Thus, the reviewers did not have the opportunity to acknowledge or comment on these revisions. Actually, we have **directly and thoroughly addressed all substantial concerns** raised in the reviews. In particular, we conducted additional experiments, strengthened the theoretical discussion, and clarified several practical aspects of the method. As a result, the manuscript has been substantially improved in terms of **experimental coverage, theoretical clarity, and practical analysis**. Detailed point-by-point responses can be found in our **General Response** and **Official Responses** to four reviewers.

Additionally, we would like to respectfully clarify a point regarding Reviewer **LqEa**’s comments. In the previous review, the reviewer strongly encouraged us to provide video evidence of deformation dynamics. However, we had already included a link to a video in our original submission. We believe this may have been unintentionally overlooked. Given the relevance of the video demonstrations to validating the effectiveness of our method, we reasonably believe that **if the reviewer had noticed it earlier**, their evaluation of our work would likely **have been more positive**.


We hope this summary will help you, as Area Chair, to assess our updated submission. Thank you very much for your time and consideration.

Best regards,

Authors of ICLR Submission 2293

---

### Meta-Review · Area_Chair_9Gav · 2026-01-06

**Summary:**

After carefully reviewing the submission, the rebuttal, and the revised manuscript, I recommend rejecting this submission. While the authors have made considerable effort to address reviewer feedback, several fundamental concerns regarding the methodological contribution, experimental scope, and practical utility of the proposed simulator remain unresolved.

**Reviewer Concerns:**

The reviewers collectively identified the following major weaknesses:

**Limited and Questionable Benefit of Full SE(n) Equivariance**: Multiple reviewers (LqEa, VqQ7) noted that the fully SE(n)-equivariant variant often underperforms compared to its non-equivariant or translation-only (R^n) counterpart, especially in short-horizon predictions. This critically undermines the paper's central claim that end-to-end equivariance is a key advantage. The authors' explanation—that stricter equivariance makes optimization harder but improves long-horizon stability—is acknowledged but not sufficiently compelling to offset the observed performance deficit in core metrics.

**Insufficient Demonstration of Scalability and Generalization**:

Multi-object Interaction: Reviewers (k92Y, VqQ7, eMaX) found the scalability demonstration weak. The extension from two to four objects requires fine-tuning and shows large errors. The work does not demonstrate the "plug-and-play" generalization or ability to handle complex, many-object (e.g., 10+) scenarios that would signify a robust and useful simulator. The authors' argument that patterns beyond four objects are not fundamentally new is speculative and unsupported.

3D Geometric Complexity: The initial experiments were criticized (k92Y) for using only simple cubes. While the authors added a cow shape experiment, the overall evaluation still lacks diversity in 3D geometry (e.g., thin structures like dominoes, standard graphics models like bunny) to convincingly demonstrate general applicability.

**Incomplete Empirical Analysis and Baselines:**

The analysis of why strong baselines like SGNN fail was initially lacking. The authors' later attribution to "discrete representation" and "large graph size" is reasonable but came only in rebuttal and feels post-hoc.

Concerns about hyperparameter sensitivity (collision radius/threshold) and computational efficiency were addressed with new experiments, which is a positive step. However, these additions do not rectify the higher-level methodological concerns.

**Unaddressed Broader Applicability Questions**: Reviewers raised pertinent questions about handling rigid-body collisions, varying material properties, and complex boundary conditions. The authors' responses largely deferred these to future work or argued they were out of scope, which limits the perceived impact and generality of the proposed framework.

**Reviewer Scores:**

While the revisions might have shifted individual scores upward, particularly for Reviewers eMaX and k92Y, the projected scores suggest the paper would likely have remained in a borderline or contentious zone. Reviewer LqEa's core criticism about the equivariant property's value remains a significant anchor. In a full discussion, these divergent views (with some reviewers potentially moving to a clear accept and others remaining at a weak reject) would have presented a challenging case for the Area Chair, heavily dependent on the weight given to methodological novelty versus empirical performance. The final decision would hinge on whether the committee judged the geometric consistency guarantees of the full SE(n) model to be a sufficiently valuable contribution, despite its sometimes inferior accuracy, to overcome the remaining reservations about practical scalability and generalization.

---

### Decision · Program_Chairs · 2026-01-26

Reject